

# The competing impacts of climate change and nutrient reductions on dissolved oxygen in Chesapeake Bay

Isaac D. Irby[1], Marjorie A. M. Friedrichs[1], Fei Da[1] and Kyle E. Hinson[1]

[1] Virginia Institute of Marine Science, College of William & Mary, Gloucester Point, VA 23062

*Correspondence to*: Isaac D. Irby (isaacirby@gmail.com) and Marjorie A. M. Friedrichs
(marjy@vims.edu)

**Abstract.** The Chesapeake Bay region is projected to experience changes in temperature, sea level, and precipitation as a result of climate change. This research uses an estuarine-watershed hydrodynamic-biogeochemical modeling system along with projected changes in temperature, freshwater flow, and sea level rise for a 2050 scenario to explore the impact climate change may have on future Chesapeake Bay dissolved oxygen (DO) concentrations and the potential success of nutrient reductions in attaining mandated estuarine water quality improvements. Results indicate that warming Bay waters will decrease oxygen solubility year-round, while also increasing oxygen utilization via respiration and remineralization, primarily impacting bottom oxygen in the spring. Rising sea level will increase the volume of the Bay, pushing coastal saline water further into the Bay. Changes in precipitation are projected to deliver higher winter and spring freshwater flow and nutrient loads, fueling increased primary production. Together, these multiple climate impacts will lower DO throughout the Chesapeake Bay and negatively impact progress towards meeting water quality standards associated with the Chesapeake Bay Total Maximum Daily Load. However, this research also shows that the potential impacts of climate change will be significantly smaller than improvements in DO expected in response to the required nutrient reductions, especially at the anoxic and hypoxic levels. Overall, increased temperature exhibits the strongest control on the change in future DO concentrations, primarily due to decreased solubility, while sea level rise is expected to exert a small positive impact and increased winter river flow is anticipated to exert a small negative impact.



## 1 Introduction

Global climate change is projected to alter the world's marine environments with coastal and estuarine systems bearing exacerbated impacts. Rising temperatures and sea levels, along with changes in precipitation patterns, have the potential to dramatically alter water quality conditions in these highly productive and increasingly human-influenced systems (Najjar et al., 2010; Altieri and Gedan, 2015). While there are multiple metrics with which to evaluate water quality, dissolved oxygen (DO)

concentrations are widely used to identify systems in distress. Large volumes of hypoxic water (generally considered to be waters with DO < 2 mg $L^{-1}$), commonly referred to as dead zones, can be found in many coastal oceans and estuaries around the world (Diaz and Rosenberg, 2008). As the climate continues to change, it is important to evaluate the impact these changes will have on DO concentrations in critical coastal environments like the Chesapeake Bay.

Climate change is generally predicted to have a net negative effect on DO in coastal waters through changes in temperature, sea level and precipitation (Boesch et al., 2007; Meier et al., 2011; Altieri and Gedan, 2015). Higher temperatures impact both the timing and rates of biological functions, while also potentially driving changes in oxygen production and consumption (Winder and Sommer, 2012). Although increased temperature is not anticipated to have a major effect on estuarine stratification, which is primarily

controlled by salinity in systems such as the Chesapeake Bay (Murphy et al., 2011), the increased temperature will act to reduce the amount of oxygen a given volume of water can hold via decreased solubility. Sea level rise (SLR) can act to increase estuarine circulation (Chua and Xu, 2014), water column stratification, residence time (Hong and Shen, 2012), and water body volume. These impacts are possibly counteractive, as increasing volume and circulation can bring in high-oxygen water from the coastal ocean,

while increased stratification inhibits downward mixing of the high-DO water from the surface waters. In addition, over much of the mid-Atlantic region annual precipitation, and thus river discharge, has been increasing (Yang et al., 2015a; Yang et al., 2015b, Tian et al., 2015). In the future, precipitation is most likely to increase most during the winter/spring and in the northern part of the region (Najjar et al., 2009; IPCC Annex I, 2013), delivering higher river flows and nutrient loads that fuel spring productivity and

produce more organic matter available for summer decomposition (Najjar et al., 2010). Changes in nutrient loading and hydrologic conditions can also alter the Bay's phytoplankton composition, changing the biomass available for eventual decomposition (Harding et al., 2015, 2016).

Compounding the complicated process of projecting future water quality conditions are nutrient management efforts such as the Chesapeake Bay 2010 Total Maximum Daily Load (TMDL; USEPA,

2010) that was developed to improve water quality conditions in the Bay by decreasing nutrient and sediment loads. These nutrient management efforts should be fully implemented by 2025 with the ultimate goal of reducing summer hypoxia (Keisman and Shenk, 2013). Examining the potential impact of climate change in light of these mandated nutrient reductions is important because the multiple impacts of climate change have the potential to render current nutrient reduction goals inadequate (Justic et al., 2007; Meier et

al., 2013; Altieri and Gedan, 2015). Furthermore, assessing the science behind climate change impacts is





critical for policies like the Chesapeake Bay TMDL that are prone to legal challenges (McCormick et al., 2017).

While much of the discussion around water quality regulations focuses on hypoxia (DO < 2 mg L$^{-1}$), studying low-DO water that encompasses concentrations greater than hypoxic levels (DO concentrations up to 5 mg L$^{-1}$) is also critical due to the impact of increases in temperature on economically important fisheries. For example, not only do temperature increases impact DO concentrations, but they also increase metabolic rates in fish. This increase causes fish to experience adverse health impacts at higher and higher DO concentrations (Portner and Knust, 2007; Vaquer-Sunyer and Duarte, 2011; Bucheister et al., 2013). Furthermore, the TMDL mandates multiple levels of minimum DO concentrations at various times and locations throughout the Chesapeake Bay (USEPA, 2010; Tango and Batiuk, 2013). While much of the regulation targets traditional hypoxia, the TMDL mandates a monthly mean DO $\geq$ 3 mg L$^{-1}$ in the deep water of the Bay to protect the survival and recruitment of Bay anchovy eggs and larvae, and a monthly mean of DO $\geq$ 5 mg L$^{-1}$ above the pycnocline to protect the growth of larval, juvenile, and adult fish and shellfish (Tango and Batiuk, 2013).

This study examines the impact of climate change on oxygen concentrations in the Chesapeake Bay by utilizing a coupled hydrodynamic-biogeochemical model that has previously been compared to other Chesapeake Bay models (Irby et al., 2016). As the EPA has stipulated a time horizon of 2025 for full nutrient reduction implementation, this research assumes that the required nutrient management strategies will be in place and limiting nutrient delivery to their full potential by 2050. With that in mind, the present study assumes the mandated nutrient reductions are implemented and employs projections of 2050 temperature, SLR, and watershed precipitation to examine the individual and combined impacts of these variables on future anoxic (< 0.2 mg L$^{-1}$), hypoxic (< 2 mg L$^{-1}$) and low-DO (2 – 5 mg L$^{-1}$) water in the Chesapeake Bay.

## 2 Methods

### 2.1 ChesROMS-ECB

The estuarine model is based on the Regional Ocean Modeling System (ROMS; Shchepetkin and McWilliams, 2005) and uses the Chesapeake Bay curvilinear horizontal grid (ChesROMS) of Xu et al. (2012) with an average wet cell resolution inside the Bay of 1.7 km. As in Feng et al. (2015), the model is configured to use the recursive MPDATA 3-D advection scheme for tracers, third-order upstream advection scheme for horizontal momentum and fourth-order centered difference for momentum in the vertical, with a 20-layer vertically stretched sigma grid. The Estuarine-Carbon-Biogeochemistry (ECB) component of the model (Feng et al., 2015) was developed originally from a continental shelf application (Hofmann et al., 2011), using dissolved organic matter cycling similar to that described in Druon et al. (2010). With only single phytoplankton and zooplankton classes and only one limiting nutrient (nitrogen), the ECB model is simpler than that employed by the Chesapeake Bay Program (Cerco et al., 2010), but is





more complex than simple dissolved oxygen models that utilize a constant oxygen consumption rate (e.g. Scully, 2010; Bever et al., 2013). ChesROMS-ECB has been previously shown to adequately resolve the spatial and temporal variability of key physical and biological variables such as temperature, salinity, nitrogen, and DO (Feng et al., 2015; Irby et al., 2016).

Before using ChesROMS-ECB to determine the impact of changes in temperature on water quality parameters, the temperature dependence of the biogeochemical formulations within the model required a careful evaluation. Several biogeochemical formulations within ChesROMS-ECB did not previously include a dependence on temperature, and temperature dependence was added as part of this study (a complete list of model changes is provided in Appendix A). For example, temperature-dependence was

introduced to the rates for maximum phytoplankton growth, zooplankton grazing/growth, nitrification, detrital solubilization, and detrital remineralization. All modifications introduce an exponential relationship between temperature and maximum rate, except for maximum phytoplankton growth. The function for phytoplankton growth is based on Lomas et al. (2002) and employs a constant growth rate below 20°C of 2.15 d$^{-1}$, with an exponential maximum growth curve for temperatures above 20°C. Remineralization of the

dissolved organic constituents previously included temperature dependence, but to ensure consistency, these rates were modified to match the Chesapeake-specific community respiration $Q_{10}$ values from Lomas et al. (2002).

An additional two changes were made to improve the light attenuation parameterization in ChesROMS-ECB. First, a minimum value of 0.6 m$^{-1}$ was applied to the diffuse attenuation coefficient,

based on model-data comparisons (Wang et al., 2009; Son and Wang, 2015). Second, the organic portion of the total suspended solids term in the light attenuation formulation of Feng et al. (2015) was multiplied by two, since carbon is generally considered to be roughly half of the total weight of organic matter.

To assess the relative skill of the revised model, the skill in reproducing water quality observations at 23 stations along the Bay (Fig. 1, Table A1) was compared to the skill of the earlier version of the model

used in Feng et al. (2015) and Irby et al. (2016). The 23 stations were assigned to four regions that are functionally delineated by salinity characteristics, with Region A representing the oligohaline, Regions B and C representing the upper and lower mesohaline (and generally the lowest DO concentrations), and Region D representing the polyhaline. The updated model retained its gross skill in terms of total root mean squared difference (Table A2) compared to the version of the model evaluated in Irby et al. (2016).

Specifically, the updated model improved bottom DO skill in Regions C and D, primarily due to the light attenuation modifications mentioned above (see Appendix A for details).

## 2.2 Chesapeake Bay Program Watershed Model

This study utilizes freshwater discharge and riverine nitrogen and sediment concentrations from the Chesapeake Bay Program's Watershed Model (version 5.3.2) that was used in the development of the

2010 TMDL (Shenk and Linker, 2013). (As in Feng et al. (2015), riverine carbon concentrations that are required as inputs to ChesROMS-ECB were obtained from the Dynamic Land Ecosystem Model (Tian et



al., 2015)). This research generally assumes that the management practices required to meet the 2010
TMDL nutrient reductions in the absence of climate change (Shenk and Linker, 2013) are fully realized by
2050; however, a brief examination of the potential impact of climate change without nutrient reduction is
also explored. Because the TMDL is based on a reference time period of 1993-1995 (USEPA, 2010), these
are the years used in this study. Fortuitously, this period includes both relatively wet years (1993, 1994)
and a dry year (1995). Simulations using the TMDL reduction in nutrient concentrations are hereafter
referred to as the TMDL scenarios while the base 1993 to 1995 simulations will hereafter be referred to as
the Base run (Table 1).

**2.3 2050 Climate Change Scenarios**

A 2050 climate change time horizon was chosen because it is far enough in the future to allow the
assumption that the TMDL nutrient reductions have been fully implemented (including nutrient transport
lag effects), while also being soon enough for relatively constrained projections of climate change impacts.
The climate change scenarios used in this research are primarily based on Coupled Model Intercomparison
Phase 5 projections for Representative Concentration Pathway (RCP) 4.5, a mid-severity future climate
scenario used in the 5[th] Assessment of the Intergovernmental Panel on Climate Change (IPCC), that
projects a peak in emissions around mid-century combined with a stabilization of radiative forcing by 2100
(IPCC Summary, 2013). It should be noted that for 2050 projections, studies have demonstrated that the
difference between RCP scenarios is smaller than the spread of individual global climate models that utilize
the RCP emission scenarios (e.g., Goberville et al., 2015). The projected regional impacts for three aspects
of climate change (temperature, SLR, and precipitation/rivers) have been included and are discussed below.

**2.3.1 Temperature**

By 2050, the Chesapeake Bay region is expected to experience air temperature increases greater
than the global average. Specifically, the IPCC projection of median annual average atmospheric
temperature increase for 2046-2065 relative to 1986-2005 for the Chesapeake Bay region is about 2°C
($\sim$0.036°C y$^{-1}$; IPCC Annex I, 2013), whereas the analogous global increase is projected to be 1.4°C
($\sim$0.025°C/y; IPCC Summary, 2013). Further research from the IPCC establishes that ocean warming tends
to be 20 to 40% lower than the rate of atmospheric warming (Collins et al., 2013). As the Chesapeake Bay
is a relatively shallow, well-mixed estuary and there has recently been an observed increase in the rate of
Chesapeake Bay warming (Ding and Elmore, 2015), this research utilizes a ratio between atmospheric and
ocean warming that is slightly lower than the open ocean range. The 1.75°C ($\sim$0.032°C y$^{-1}$) increase in Bay
water temperature for 2050 relative to the mid-1990s used in this study (Table 1) is higher than the $\sim$0.02°C
y$^{-1}$ observed Chesapeake Bay warming between 1949 and 2002 (Preston, 2004). However, Preston (2004)
found evidence of increased warming in the late 1990s. The rate of warming used in this analysis is also
consistent with projected increases by the end of the century from downscaled global climate models for
the Bay (Muhling et al., 2017). It is also slightly lower than the warming estimated using a high resolution
climate model (CM2.6; Saba et al., 2015) for the location of the ChesROMS open boundary (2.6°C, Saba



pers. comm.), and less than the average satellite-derived rate of Bay surface water warming of 0.005-
0.175°C/y from 1984 to 2007 (Ding and Elmore, 2015).

The 1.75°C water temperature increase was applied uniformly across time and space to
biogeochemical process and oxygen solubility throughout the Bay, but the temperature increase was not
applied to other physical properties or processes, such as water density gradients or meteorological forcing.
Thus, increased temperature affects do not impact stratification or other physical dynamics of the Bay
within the model. This approach implicitly assumes that the Bay is shallow enough that climatic warming
will occur uniformly over time. Supporting this assumption, Preston (2004) found that the surface and
subsurface waters of the Bay warmed at relatively similar rates, even finding that, on average, the
subsurface waters warmed slightly faster than surface waters. In addition, recent trends in the
intensification of early summer stratification have been found not to be due to water column temperature
changes, but rather are primarily due to changes in salinity as a result of SLR and altered freshwater inputs
(Murphy et al., 2011). Changes in salinity along the ChesROMS open boundary on the continental shelf
between the 1990s and 2050 have been computed by Saba et al. (2016) to be very minor (~0.2 psu) and are
thus not considered here. The temperature increase scenario will hereafter be referred to as the
TMDL+tempCC scenario since the increase in temperature is applied to the TMDL nutrient scenario (Table
1).

**2.3.2 Sea Level Rise (SLR)**

The Chesapeake Bay is expected to incur a greater increase in sea level than the global average,
and the Bay has experienced a recent acceleration in SLR, as has most of the Mid-Atlantic coast (Sallenger
et al., 2012). Boon and Mitchell (2015) found a roughly 0.1m increase in sea level in Norfolk, Virginia
between 1993 and 2014. Assuming a linear extrapolation of that rate (~5mm y$^{-1}$), by 2050 Norfolk would
expect a SLR of 0.3m relative to the mid-1990s. However, the linear extrapolation ignores the projected,
and recently observed, acceleration in SLR. Incorporating anticipated acceleration, Boon and Mitchell
(2015) estimate an average increase in SLR by 2050 of ~0.5m in the Chesapeake Bay relative to the
relative mean sea level between 1969-2014. Using downscaled global models, Sweet et al. (2017) estimate
a similar SLR in the Mid-Atlantic for 2050 under an intermediate emissions scenario. This research
assumes a 2050 SLR of 0.5m (~9mm y$^{-1}$) relative to the mid-1990s, which is consistent with these recent
regional projections (Boon and Mitchell, 2015; Sweet et al., 2017).

Model implementation of SLR follows that of Hong and Shen (2012). The 0.5m increase was
added to the free water surface layer at the outer boundary of the model grid, along the continental shelf.
The vertical grid stretching parameters were not altered and the simulation required less than six months for
the Bay to equilibrate to the increased sea level. The SLR scenario will hereafter be referred to as the
TMDL+slrCC scenario since the 0.5m increase is applied to the TMDL scenario (Table 1).



### 2.3.3 River Flow


The Chesapeake Bay watershed spans a range of projected precipitation changes with the southern portion of the watershed expected to experience a lower intensity change than the northern portion of the watershed, complicating projections of precipitation change, and as a result, river flow (Najjar et al., 2009). While precipitation exerts a first order control on river flow, the projected changes in river flow derived from a watershed model is also greatly influenced by the choice of potential evapotranspiration (PET) parameterization. The PET parameterization used here for the climate change experiments is based on the Hargreaves-Samani equation (Hargreaves and Samani, 1982). The Hargreaves-Samani equation is a


simplistic representation of evapotranspiration dynamics as it only explicitly accounts for temperature, but does not include advective processes and only implicitly represents relative humidity by including the difference in maximum and minimum temperature. In addition, the 2050 Watershed Model projections include a parameterization for increased stomatal resistance due to elevated $CO_2$.

The river flow projections used here are derived from a watershed simulation that incorporated


downscaled precipitation and temperature estimates for the RCP4.5 scenario from 32 Global Climate Model realizations. All model results used were first downscaled to a 1/8° resolution over the Chesapeake Bay watershed, using a bias-corrected spatial disaggregation (Reclamation, 2013). The 32 model results for both precipitation and temperature were compared for each month, and the median model estimate was chosen to represent the change that would be applied to watershed model inputs. Changes in rainfall were


also distributed unequally among different precipitation events throughout the months in the simulation period in order to increase the intensity based on estimates provided by Groisman et al. (2004). Overall, the changes in precipitation applied to the Watershed Model inputs resulted in greater precipitation and runoff, especially in the winter and spring months. However, the warmer temperatures throughout the year mitigated some of these increases via increased rates of evapotranspiration.


From these Watershed Model results, the ratio of monthly freshwater delivery to the Bay for the climate change scenario relative to the base case was calculated for the Susquehanna River (Table 2), and was applied to all rivers in ChesROMS-ECB. This is a reasonable approach given that the Susquehanna watershed accounts for > 80% of the Bay watershed area that drains directly to the main stem and is the primary source of the nutrients that drive the summer hypoxic region of the Bay between the Patapsco


River in the north and the Rappahannock River in the south (Hagy et al., 2004). Overall, there is an increase in river flow applied to the model (Table 2). This increase in river flow results in both an increase in freshwater discharge and an increase in nutrient delivery. The combined impact of increased freshwater flow and nutrient loads will hereafter be referred to as the TMDL+riverCC scenario (Table 1).

### 2.3.4 Combined Climate Change Scenario


A final scenario that combines all three of the climate change impacts was run for both nutrient cases, i.e. the TMDL scenario (reduced nutrients) and the Base run (realistic nutrients). These scenarios



will hereafter be referred to as the TMDL+allCC and Base+allCC scenarios respectively, since the combined impact of all climate change variables (temperature, SLR, and rivers) was applied (Table 1).

**2.4 Dissolved Oxygen Analysis**

Hypoxic volume is a commonly used metric to quantify the amount of water that experiences a given level of DO concentration over a specific time (e.g. Murphy et al., 2011; Bever et al., 2013). In this study, two metrics related to hypoxic volume are computed: cumulative hypoxic volume (CHV) and hypoxic duration (HD). CHV is calculated as the sum of each day's hypoxic volume over a year (Bever et al., 2013), and HD is computed as the number of days that have a hypoxic volume greater than 1 km$^3$.

While traditional DO concentration levels of hypoxia (< 2 mg L$^{-1}$) and anoxia (< 0.2 mg L$^{-1}$) are examined, this research also considers impacts of low-DO, defined here as DO < 5 mg L$^{-1}$. This level is consistent with the highest DO concentrations stipulated in the Chesapeake Bay water quality standards (USEPA, 2010) and is a conservative upper bound on DO concentration found to initiate stress on marine fish (Vaquer-Sunyer and Duarte, 2008; Buchheister et al., 2013).

**3 Results**

The impact of nutrient reduction on bottom DO concentrations is greater than that of climate change (Fig. 2). The reduction of nutrients (between the Base run and TMDL scenario) causes a general increase in bottom DO concentrations. This impact is largest during the drawdown of bottom oxygen in the

spring (April – June), dampens during the course of the summer, and is lowest in winter (Dec – Feb). In Region B, the region of the Bay where oxygen concentrations are lowest and most persistent, this impact is strongest in the driest year (1995), during which the increase in bottom DO exceeds 2.5 mg L$^{-1}$. In 1993 and 1994 the bottom DO increase is only around 1.5 mg L$^{-1}$ (Fig. 2). In contrast Region C, encompassing the southern extent of the hypoxic zone, experiences a greater increase in spring bottom DO than Region B in

the wet years (>2 mg L$^{-1}$ in 1993 and 1994) and a smaller increase in the dry year (~1.5 mg L$^{-1}$ in 1995).

Climate change has a smaller effect on bottom DO concentrations than the TMDL nutrient reductions. Climate change has almost no impact on bottom DO during the peak of summer when bottom DO concentrations are the lowest (near zero). In the Base run (realistic nutrient inputs), the effect of climate change on spring bottom DO is a decrease of ~0.6 mg L$^{-1}$ and ~0.8 mg L$^{-1}$ in Regions B and C respectively.

Climate change impacts bottom DO similarly in the TMDL scenario, with reductions in spring bottom DO of ~0.5 mg L$^{-1}$ in both Regions B and C (Fig. 2). In both regions, these reductions in bottom DO are similar in all three years.

Of the three climate factors considered (temperature, SLR and river flow), temperature had the largest impact on bottom DO. As a result, the TMDL+allCC scenario is most similar to the

TMDL+tempCC scenario (Fig. 3). In Region B, the TMDL+slrCC and the TMDL+riverCC scenarios have a smaller impact on bottom DO during the wet years of 1993 and 1994 than during the dry year of 1995. The opposite occurs in Region C, with the TMDL+slrCC and the TMDL+riverCC scenarios having a larger





impact on bottom DO during the wet years of 1993 and 1994 than during the dry year of 1995. In both regions, the impact of SLR generally increases bottom DO during the spring and summer, while changes in

the rivers (increased seasonality and nutrient load) suppress DO. These two essentially equal and opposite effects largely cancel each other out (Fig. 3).

Although temperature had the largest impact on bottom DO in each of the four regions considered, the magnitude of the individual impacts of the climate change scenarios differed by region (Table 3). Specifically, in the TMDL+allCC scenario, bottom DO decreased compared to the TDML+noCC run in all

four regions, with Region A exhibiting the highest total average change ($-0.58$ mg L$^{-1}$) and the other three regions all exhibiting an average change of roughly $-0.4$ mg L$^{-1}$ (Table 3). This is primarily due to the large decreases in bottom DO in the TMDL+slrCC scenario in Region A ($-0.21$ mg L$^{-1}$), relative to the small (mostly positive) impacts due to sea level rise in the other regions. Overall, the impact of all three of the climate change factors (temperature, SLR, river flow) is nearly linearly additive (Table 3).

The CHV for all of the TMDL scenarios (both with and without climate change) is less than the CHV from the Base run without climate change (Fig. 4). This pattern holds true for all six DO levels examined ($< 0.2$ mg L$^{-1}$ to $< 5$ mg L$^{-1}$). At each DO level, the CHV for the dry year (1995) is much less than for the wet years (1993 and 1994) for each TMDL scenario. Furthermore, the CHV for the TMDL scenarios in the wet years is generally higher than the CHV from the Base run for the dry year (Fig. 4). The

CHV in the TMDL+slrCC and TMDL+riverCC scenarios tend to track closely to the TMDL+noCC scenario, while the TMDL+tempCC scenario is most similar to the TMDL+allCC scenario (Fig. 4), as was also the case for bottom DO (Fig. 3).

The percent change in CHV relative to the progress, or gains, made in CHV by applying the TMDL nutrient reductions varies across DO level and by scenario (Fig. 5). In general, the TMDL+slrCC

scenario resulted in a ~0-10% increase in the improvement made by the TMDL scenario (here, an increase of gains is actually a decrease in CHV) across all DO levels and all years. In contrast, the TMDL+riverCC and TMDL+tempCC scenarios resulted in a degradation of the system, compared to the TMDL+noCC scenario. The TMDL+riverCC scenario consistently causes a loss of ~0-5% of the gains, with slightly larger losses in 1994 and 1995 at higher DO levels. The TMDL+riverCC scenario combines two separate,

but linked, climate change impacts: increased freshwater flow (particularly in the winter) and increased nutrient loads (as a result of increased freshwater flow). While not shown, separate experiments isolating the impacts of flow and load demonstrated that the increase in nutrient load, rather than the increase in freshwater flow, caused the increase in CHV in the TMDL+riverCC scenario. The TMDL+tempCC scenario was the strongest function of DO level, with a relatively small loss of ~5% at the $< 0.2$ mg L$^{-1}$

level and a large ~40% loss at the $< 5$ mg L$^{-1}$ level (Fig. 5). The combined effect of climate change (TMDL+allCC) was a net increase in CHV of more than 50% over the TMDL+noCC scenario in the wet years of 1993 and 1994 for DO $< 5$ mg L$^{-1}$, and a corresponding 40% increase in CHV for the dry year of 1995 (Fig. 5).



As shown above, increased temperature generally maintains the greatest control on the
TMDL+allCC scenario (Figs. 4). The impact of temperature on DO in this analysis is due to two factors:
chemical solubility and biological oxygen demand. To isolate these impacts, the differences in modeled DO
computed with and without warming are computed considering only solubility effects and considering both
solubility and biological oxygen demand (Fig. 6). Since oxygen saturation is more sensitive to changes in
temperature at low temperatures, there is a larger change in DO as a result of changes in solubility during
the winter than during the summer, even though the change in temperature is constant in time. Deviations
from the change in DO due to solubility can be attributed to changes in biological oxygen demand, and can
be estimated by comparing the simulation assuming only solubility impacts (red line in Fig. 6) with the
simulation assuming temperature changes affect both solubility and biological oxygen demand (black line
in Fig. 6). Overall, 65-85% of the change in DO between the TMDL+tempCC scenario compared to the
TMDL+noCC scenario is a result of temperature's impact on solubility. The impact of biological oxygen
demand is consistently negative at depth during the spring and early summer, enhancing the initiation of
hypoxic conditions (Fig. 6b).

In terms of the number of days that the Bay experiences hypoxic and low-oxygen conditions each
year, climate change reduces the positive impact of the nutrient reduction (Fig. 7). While there is a large
decrease in hypoxic duration resulting from the nutrient reduction, the TMDL+allCC scenario demonstrates
that when climate change is included all levels of low-DO and hypoxia initiate an average of ~7 days
earlier. This trend is not evident in the cessation of hypoxia and low-DO, i.e. climate change does not
necessarily cause hypoxia to last later in the year. While all three years exhibit a similar pattern and
timeline of cessation of low-DO with < 0.2 mg L$^{-1}$ ceasing 3-4 months before < 5 mg L$^{-1}$, each year is
different in terms of initiation timing. In 1993 for the Base+noCC run, all levels of DO initiate within 2
weeks of each other. This timing holds true for the TMDL scenarios as well, but with anoxia lagging
behind. In 1994 in the Base+noCC run, there is a steady progression from low-DO to anoxia over ~6
weeks. In the TMDL scenarios, that is extended to ~3 months. In 1995, the TMDL nutrient reduction
results in no DO < 1 mg L$^{-1}$ and significantly delays the onset of low-DO by up to ~3 months compared to
the Base run.

Nutrient reduction primarily reduces the horizontal extent of the hypoxic zone (Fig. 8). Examining
a south-north transect along the main stem of the Bay for July 1$^{st}$, 1993 (Fig. 8a,c) and 1995 (Fig. 8b,d)
reveals that nutrient reduction acts to compress the southern extent of the hypoxic zone more than the
northern extent. One similarity between all four subplots (a-d) is the vertical extent of the low-oxygen
waters, which are capped by the pycnocline at ~ 5m depth. As expected, the extent and severity of anoxia
and hypoxia on July 1$^{st}$ is much greater than the summer (May-September) average for both the
Base+noCC run and TMDL+noCC scenario (Fig 8e-h). In general, the impact of climate change is greater
in the dry year (1995; Fig. 8j,l) than in the wet year (1993; Fig. 8i,k). The location of the greatest
magnitude change is near the pycnocline depth (Fig. 8i,j) but the location of greatest percent change is
below the pycnocline (Fig. 8k,l).



The climate change scenarios cause a larger volume of the Bay to experience low-DO concentrations in both wet and dry years and under both the Base+allCC and TMDL+allCC scenarios (Fig. 9). While climate change does not greatly exacerbate the volume of the Bay that experiences anoxic and hypoxic conditions, climate change increases the percent of the Bay experiencing conditions of DO < 5mg L$^{-1}$ by up to ~6 %, regardless of whether or not the TMDL nutrient reductions have occurred. Similarly, regardless of whether or not climate change occurs, the volume of the Bay experiencing low-DO under nutrient reduction is ~10% lower than that in the 1993-1995 Base run nutrient conditions. Overall, the dry year (1995) results in ~30-50% as much of the Bay experiencing low-DO and hypoxic waters as compared to the wet years (1993, 1994).

**4 Discussion**

**4.1 How will Chesapeake Bay DO concentrations change in the future as a result of climate change?**

- By 2050 low-DO conditions can be expected to begin about one week earlier due to climate change, with increases in volume and extent being largest at the margins and at the southern extent of the hypoxic zone. Significant impacts will be felt on water with DO concentrations in the range of 2-5mg L$^{-1}$, and not only on hypoxic waters (DO < 2mg L$^{-1}$).

The most consistent impact across all levels of low-DO waters due to climate change is an earlier onset of hypoxic and low-DO conditions by an average of ~7 days. While an earlier onset was identified, there was no trend in the cessation of hypoxic and low-DO conditions, with climate change sometimes causing an earlier and sometimes a later cessation. Furthermore, an earlier onset of conditions is projected to occur under both nutrient-reduced and nutrient-replete futures. The pattern of earlier onset is primarily due to the additive impacts of an increase in spring biological oxygen utilization at depth and decreased solubility, both the result of the increase in temperature (Fig. 6). An analysis of climate change impact on DO of an estuarine tributary of the Chesapeake Bay similarly found that hypoxic duration is likely to be extended in the future (Lake and Brush, 2015).

In terms of a change in the volume of low-DO waters, the relative impact of climate change increases with DO concentration (Figs. 4, 5). The improvements due to the nutrient reductions are reduced by climate change, ranging from ~5% for DO < 0.2 mg L$^{-1}$ to ~45% for DO < 5 mg L$^{-1}$. The difference between impact at anoxic levels versus waters with DO of 3 - 5 mg L$^{-1}$ is accentuated during the dry year of 1995 due to the fact that the nutrient reductions result in no modeled DO < 1 mg L$^{-1}$ during this year (Fig. 7), regardless of whether or not climate change is occurring. Even assuming realistic 1995 nutrient inputs, the volume and duration of anoxia under climate change in 1995 is very small.

Throughout the water column, the greatest change in DO will be at the edges of the low-DO and hypoxic zones, particularly at the southern and vertical extents (Fig. 8). Conversely, the smallest changes will occur in the anoxic waters where DO cannot be decreased further (Fig. 8). As hypoxia is capped by the





pycnocline (Irby et al., 2016), the magnitude of DO change ($\sim 0.5$ mg L$^{-1}$) is not great enough to extend low-DO conditions to the DO-replete surface waters. Laterally, the largest changes in bottom DO will be in the southern extent of hypoxia and the degree of east-west compression along the main stem of the Bay. Such a change would be likely to detrimentally impact demersal fish and shellfish communities along the

shallow flanks of the Bay and its tributaries.

**4.2 How will the individual impacts of climate change (increased temperatures, SLR, river flow) affect DO concentrations in the Chesapeake Bay?**

- The combined impacts of climate change will cause reduced DO concentrations in 2050, with increased water temperatures being the strongest driver of this change.


In examining the individual impacts of projected temperature, SLR, and river flow in 2050 on Chesapeake Bay DO concentrations, temperature exhibits a large negative impact, river flow exhibits a small negative impact, and SLR exhibits a mixed impact depending on region but is generally positive (Figs. 4, 5; Table 3). The large impact of increased temperature on DO in light of nutrient reduction is

consistent with other modeling research focused on the York River estuary, a tributary of the Chesapeake Bay (Lake and Brush, 2015). The present research demonstrates the importance of temperature on solubility, as the annual average impact of temperature on oxygen saturation outpaced the impact of temperature on biological functions on average by roughly 2:1 in the region of the Bay that experiences hypoxia (Fig. 6). This ratio is decreased to roughly 1:1 during the spring/early summer drawdown of

bottom DO in the main stem channel (Fig. 6). Murphy et al. (2011) similarly found that increased respiration due to increased temperature potentially plays a smaller role on changes in hypoxia than the physical and chemical changes. However, it is possible that as temperature continues to increase, the ratio of impact between solubility and biological oxygen demand may shift toward a greater influence by biological oxygen demand. This is because the additional impact of further changes in solubility will

decrease as temperatures continue to rise, while biological respiration at depth and production at the surface may continue to steadily rise with increasingly warmer temperatures.

Both SLR and changes in river flow exert their greatest relative impact during the driest year considered (1995). The increase in winter precipitation will deliver both increased freshwater flow and increased nutrient loads and accounts for a larger percentage of the overall change in DO during the dry

year of 1995 because the low-flow conditions cause the Bay to be more sensitive to changes in freshwater flow and nutrient loading. SLR also exhibits its greatest influence during 1995, causing a decrease in CHV resulting from an increase in the flux of high-DO water from the shelf and an overall increase in Bay volume acting to reduce the unit consumption of DO per volume given a consistent loading of organic matter. The larger impact of SLR during dry years is consistent with a study from the Delaware Bay

showing that high flow dampens the salinity intrusion that results from SLR (Ross et al., 2015) and with a




study in San Francisco Bay finding that the impact of SLR is limited under high flow conditions (Chua and Xu, 2014).

**4.3 How might climate change impact the success of the 2010 TMDL nutrient reductions?**

- Climate change may cause the 2010 TMDL nutrient reductions to be insufficient to meet the
required water quality improvements in the Chesapeake Bay.

This research demonstrates that the improvements in Chesapeake Bay water quality due to the TMDL nutrient reductions are much greater than the deleterious impacts of 2050 climate change; however, results also indicate that by 2050 climate change will likely decrease oxygen levels and increase both
hypoxic volume and hypoxic duration. Because some locations in the Bay barely pass TMDL standards and others require special allowances to meet the standards (Irby and Friedrichs, in revision), even these small increases in anoxic and hypoxic conditions can cause locations that previously passed the water quality standards to fail under a changing climate. The DO minima in the TMDL regulations are based on both space and time criteria. Although the spatial dimension may not be greatly impacted at the anoxic and
hypoxic levels, this research suggests that the temporal dimension will be. This could cause locations in the Bay that are currently projected to pass the minimum standards to fail them in light of climate change, simply due to an extension of the hypoxic season without an expansion of hypoxic volume.

With increased temperature being the primary cause of the impact of climate change on DO concentrations, it is important to consider other potential impacts increased temperature may have on the
ecosystem in the context of the success of the TMDL nutrient reductions. Temperature increases in the Chesapeake Bay are anticipated to produce temperatures outside of previously observed extremes (Muhling et al., 2017), lending increased pertinence to understanding the impact of temperature changes on meeting water quality goals. In light of this, the impact on the TMDL of a decrease in oxygen concentrations due to climate change should be viewed in conjunction with the impact increased temperature is likely to have on
the species upon which the DO levels in the TMDL nutrient reductions were predicated. Multiple studies have established that increasing water temperature increases metabolic rates in fish that cause them to experience negative health impacts at higher DO concentrations than they do at lower temperatures (Breitburg, 2002; Portner and Lanning, 2009; Lapointe et al., 2014). Due to those compounding impacts and the large role temperature is expected to play in regulating future DO, it may be prudent for the TMDL
to elevate the mandated minimum DO levels in an effort to protect target species. If this occurred, the impacts of climate change would likely cause an even larger failure rate of TMDL standards.

**4.4 How will climate change impact DO if the TMDL nutrient reductions are not met?**

- Although the relative impact of climate change is similar on a reduced nutrient future and a high nutrient future, the degree of interannual variability in hypoxia may change in a reduced versus
high nutrient future due to differences in the responses of oxygen to fluctuations between dry and wet years.





The relative impact of climate change on a reduced nutrient versus a high nutrient future is similar in terms of hypoxic volume and duration. In both a low and high nutrient future, the percent of the Bay that experiences a given DO level is increased with climate change (Fig. 9). Furthermore, in both cases, the impact of climate change at low-DO concentrations ($< 5$ mg L$^{-1}$) is greater than that at hypoxic levels ($< 2$ mg L$^{-1}$). In terms of relative change in DO along the main stem of the Bay, a high nutrient future is expected to experience a higher (~9-15%) change in DO concentration than a low nutrient future (~6-9%), with the largest changes in both cases occurring at the southern end of the hypoxic zone (Fig. 8).

The largest potential ecological difference between the two futures is in the dry year of 1995. In this year TMDL scenarios exhibited no anoxia in the Bay, regardless of whether or not climate change was occurring. This suggests that during dry years, when the nutrient reduction may be sufficient to alleviate anoxia, climate change impacts may not be large enough to overcome the hysteric or threshold level of DO initiation similar to what has been observed with hypoxic responses to nutrient loading (Kemp et al., 2009). It may seem counterintuitive, but this suggests that the interannual variability of anoxic conditions may be exacerbated in a future with nutrient reduction because the interannual percent change in anoxic conditions will be relative to ~0% in the very dry years. Because of this, when climate change is added to the TMDL nutrient reductions, there is likely to be greater disparity in terms of anoxic volume between wet and dry years. Further intensifying the difference between wet and dry years is the potential impact of nutrient storage in the watershed during dry years that is delivered to the Bay in a successive wet year, amplifying hypoxia and anoxia (Lee et al., 2016).

**4.5 Methodological limitations**

This research is a first order look at the potential impacts that changes in climate may have on the efficacy of nutrient reduction efforts in the Chesapeake Bay; however, more robust examinations of the problem are needed in order to adequately aid in the regulatory decision making process going forward. As the present research has identified increased temperature as the largest contributor to changes in DO, future efforts should work to incorporate the impact of increased air temperature and changes in meteorological forcing on the air-sea interface and Bay hydrodynamics. In addition, increased stream temperatures will likely need to be accounted for, as there is evidence that the current rates of Bay warming cannot be fully explained by the observed increase in regional air temperatures (Ding and Elmore, 2015). Projections of future precipitation indicate changes in storm intensification and extreme events that could have dramatic effects on nutrient delivery to the Bay (Sinha et al., 2017), and thus these should be considered in future work as well. Finally, the atmospheric wind field will likely change in the future. Although there is significantly uncertainty associated with future wind projections, the strong impact of wind on hypoxia in the Chesapeake Bay makes this an important issue to better understand.

Due to the uncertainty in projected changes in temperature, river flow, and SLR, assessing the sensitivity of DO to multiple estimates of climate change will be important. This research establishes that




the increase in temperature has the strongest control on DO, but that does not mean that DO concentrations
are most sensitive to the bounds of potential 2050 temperature changes. While the high computational

expense of running multiple sensitivity tests through complex coupled hydrodynamic-biogeochemical
models can be prohibitive, establishing a range of uncertainty is critical to informed adaptive management
decision-making.

Additional limitations are related to timing. For example, the present research assumes a
discontinuity between the reduction of nutrients and the changes in climate. This is an unrealistic

assumption because the nutrient reductions and climate change will continue to occur contemporaneously.
These changes are also not immediate but manifest over time in a continuously evolving environment. In
addition, the current approach simply identifies the potential ramifications of climate change on nutrient
reduction efforts but does not establish a timeline for the water quality changes as a result of nutrient
reductions to occur. This means that climate change has the potential to further limit the effectiveness of

nutrient reduction efforts because the impacts of climate change may be more immediate than the impacts
of nutrient reduction. To address these limitations, an effort to conduct a continuous simulation from 2015
– 2050 including both gradual implementation of the nutrient reductions and climate change impacts is
currently underway.

### 5 Conclusions


Overall, the most striking result of this research is that the potential impact of climate change in
2050 is much smaller than the impact of the 2010 TMDL nutrient reductions, particularly at anoxic and
hypoxic levels. However, the decrease in DO concentrations resulting from the combined impacts of
climate change may cause portions of the Bay that currently meet mandated water quality standards to fail

them in the future. At the most stringent DO standards, this is primarily due to an increase in hypoxic
duration rather than hypoxic volume, as under climate change, the onset of hypoxic conditions is projected
to initiate ~7 days earlier on average across all DO concentrations $0.2 – 5$ mg L$^{-1}$.

Changes in DO as a result of the increase in temperature dominate the combined climate change
impact. While the influence of solubility on DO concentrations is the primary control on decreased DO

throughout the year, the impact of increased biological oxygen demand is most prevalent at the bottom in
the spring to early summer, contributing to the earlier initiation of hypoxic conditions. The impact of
temperature is likely to affect low-oxygen tolerance of higher trophic levels as well by increasing metabolic
rates, making species less tolerant at higher DO levels. This may result in the DO minimums mandated in
the water quality standards to be insufficient to protect key species even if the current goals are met.

Both sea level rise and changes in river flow exert a greater influence on change in DO during dry,
low streamflow years. Changes in river flow are likely to deliver higher freshwater flows during the winter
and spring that will both deliver higher nutrient loads and increase estuarine circulation. These two effects
impact DO concentrations oppositely, with higher loads resulting in more organic matter being available
for decomposition and increased estuarine circulation delivering more oxygen-rich ocean water: however,





the impact of increased loads outcompetes the greater circulation. Sea level rise exerts the only net positive
impact of climate change on DO concentrations, increasing the effectiveness of the TMDL nutrient
reductions by ~5% in the mesohaline. However, this positive impact is undermined by the large negative
impact of temperature.

The relative effects of climate change are similar whether the TMDL nutrient reductions are
achieved or not. In both cases, there is a slight increase in anoxic conditions, and the relative impact of
climate change intensifies at higher DO concentrations (3 - 5 mg L$^{-1}$). The impact of the nutrient reductions
on dry years is accentuated compared to the 'business as usual' dry years due to the greater moderating
influence sea level rise exerts during low-flow conditions. This results in anoxic and hypoxic conditions to
be depressed with nutrient reduction plus climate change in the dry year of 1995, but not when climate
change is combined with no nutrient reduction.

Overall, this study demonstrates that climate change has the potential to limit the effectiveness of
future management actions aimed at reducing nutrient inputs to the Chesapeake Bay. However, the
negative impacts of climate change are smaller than the positive impacts resulting from the mandated
nutrient reductions. Given that this analysis only considers a 2050 time horizon and climate impacts are
expected to intensify with time, it is critical to continue to examine how the Bay may evolve in the future.



**Acknowledgements**



This paper is the result of research funded in part by NOAA's National Centers for Coastal Ocean
Science under award NA16NOS4780207 to the Virginia Institute of Marine Science (VIMS) and by
NOAA's U.S. Integrated Ocean Observing System Program Office as a subcontract to VIMS under award
NA13NOS0120139 to the Southeastern University Research Association. Chesapeake Bay Program
Watershed Model output was provided by G. Shenk and R. Tian. Thank you to C. Hershner, R. Hood, R.
Najjar and C. Friedrichs, for comments on an initial version of this manuscript. This work was performed
in part using computing facilities at the College of William and Mary, which were provided by
contributions from the National Science Foundation, the Commonwealth of Virginia Equipment Trust
Fund, and the Office of Naval Research. This is Virginia Institute of Marine Science contribution #####
and CHRP contribution number #####.



**Tables**

**Table 1** Scenario definitions.

| Scenario | Nutrients | Climate Change |
|---|---|---|
| Base+noCC | Realistic 1993 – 1995 nutrients | None |
| TMDL+noCC | TMDL nutrient reductions | None |
| TMDL+riverCC | TMDL nutrient reductions | River change only (Table 2) |
| TMDL+tempCC | TMDL nutrient reductions | 1.75°C increase |
| TMDL+slrCC | TMDL nutrient reductions | 0.5m increase in sea level |
| TMDL+allCC | TMDL nutrient reductions | All three above changes |
| Base+allCC | Realistic 1993 – 1995 nutrients | All three above changes |














**Table 2** Monthly freshwater discharge fractional change factor used for the TMDL+riverCC, TMDL+allCC, and Base+allCC scenarios, calculated as the ratio between the freshwater inputs in 2050 divided by the freshwater inputs in the Base Run.

| Month | Freshwater change factor* |
|---|---|
| January | 1.165 |
| February | 1.168 |
| March | 1.035 |
| April | 0.964 |
| May | 1.034 |
| June | 1.015 |
| July | 0.965 |
| August | 1.042 |
| September | 0.986 |
| October | 0.984 |
| November | 1.093 |
| December | 1.158 |













**Table 3** Average change in bottom DO (mg L$^{-1}$) relative to the TMDL+noCC run for each scenario and region.

| Scenario | Region A | Region B | Region C | Region D |
|---|---|---|---|---|
| TMDL+allCC | -0.58 | -0.37 | -0.44 | -0.44 |
| TMDL+slrCC | -0.21 | 0.09 | 0.04 | -0.04 |
| TMDL+riverCC | -0.01 | -0.05 | -0.03 | -0.01 |
| TMDL+tempCC | -0.36 | -0.40 | -0.44 | -0.38 |
| Additive impact of slrCC+riverCC+tempCC | -0.58 | -0.36 | -0.43 | -0.43 |














**Table 4** Percent* of 3-year average bottom DO change as a result of the temperature experiment due to solubility for each region at the surface and bottom of the water column.

| Region | Surface | Bottom |
|--------|---------|--------|
| A | 75% | 75% |
| B | 72% | 66% |
| C | 77% | 69% |
| D | 85% | 79% |


*Percent calculated as the expected change in bottom DO as predicted by solubility divided by the modeled change in bottom DO.





**Figures**

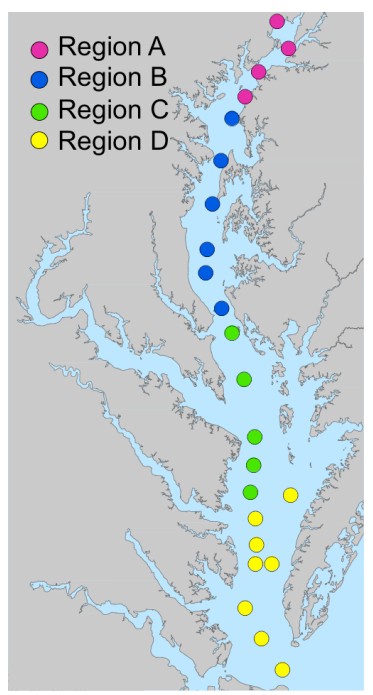


**Figure 1: Map of the Chesapeake Bay with water quality monitoring stations (Table A1) identified by region, based primarily on salinity. A: oligohaline, B & C: upper & lower mesohaline, D: polyhaline.**







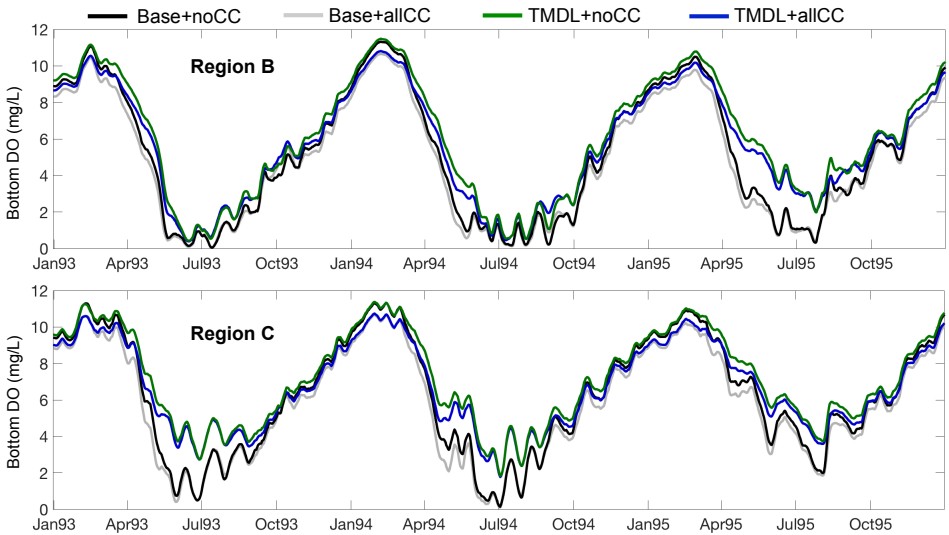


**Figure 2: Time series (7-day running mean) of bottom DO with and without nutrient reductions (TMDL vs. Base) and with and without climate change (allCC vs. noCC), for the average of the stations in the (top panel) upper mesohaline Region B and (bottom panel) lower mesohaline Region C.**










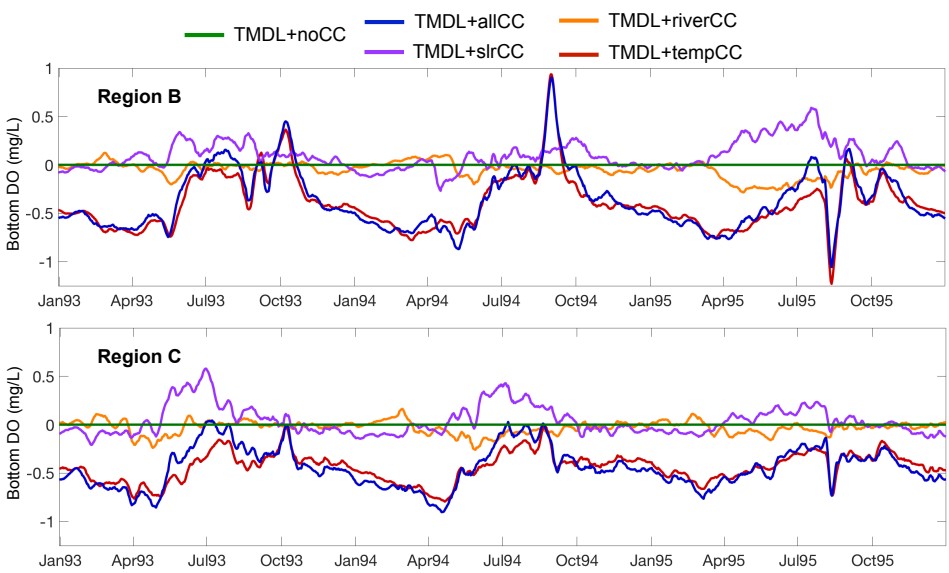

**Figure 3: Time series (7-day running mean) of the change in bottom DO between the TMDL climate change and no climate change scenarios for the average of the (top panel) upper mesohaline Region B and (bottom panel) lower mesohaline Region C.**





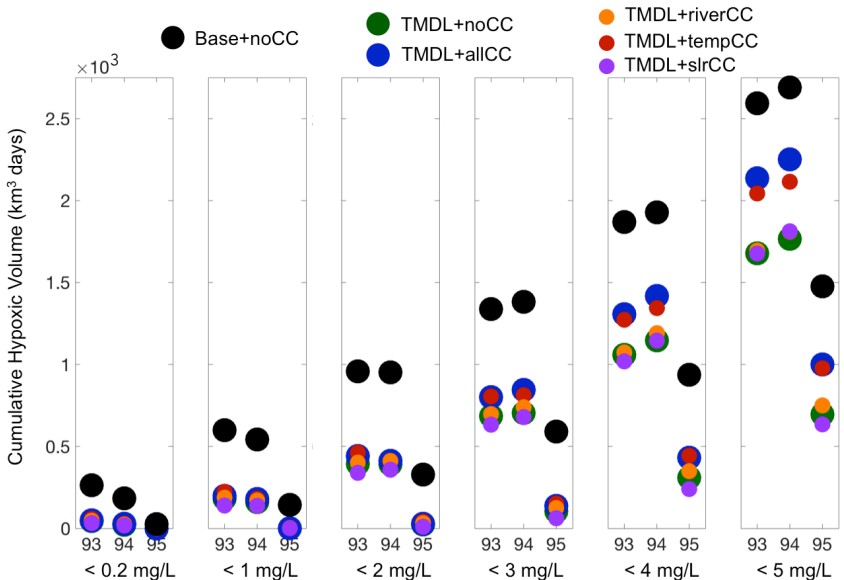

**Figure 4: Cumulative hypoxic volume for six ranges of DO concentrations, for each of the study years and each of the scenarios (colored circles).**









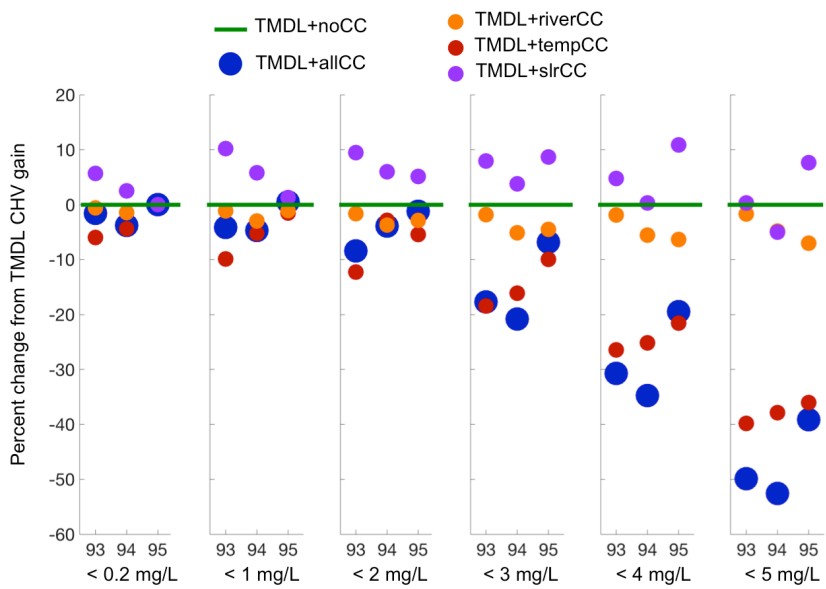

**Figure 5: Percent change due to climate change, relative to the improvement in CHV between the TMDL+noCC scenario and Base+noCC run. Percent change in CHV gain is defined as: (TMDL+xx – TMDL+noCC)/(TMDL+noCC – Base run+noCC).**









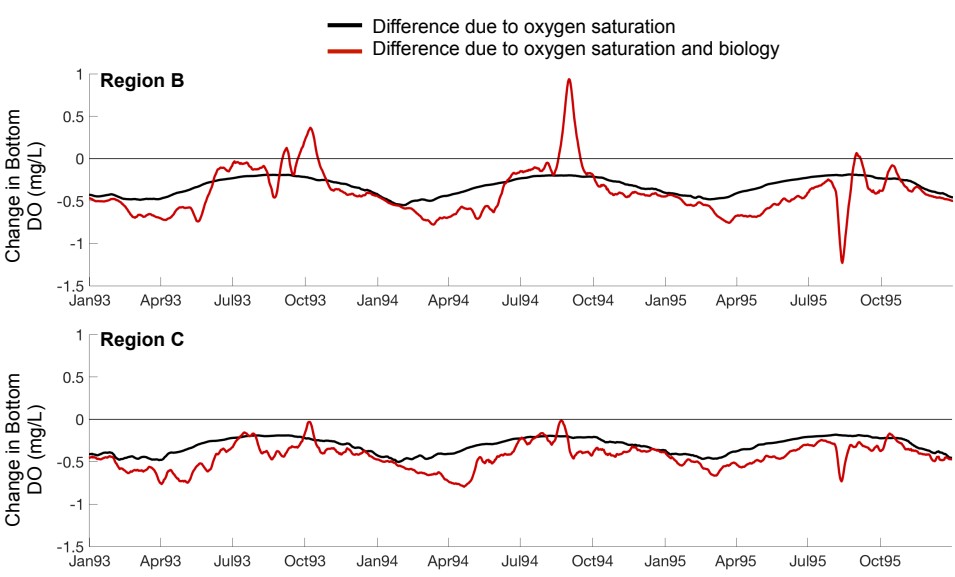


**Figure 6: DO differences due to climate change (between the TMDL+noCC and TMDL+tempCC scenarios) averaged for (top panel) the stations in Region B and (bottom panel) the stations in Region C. The black lines are the average change expected if only solubility was impacted by an increase in temperature. The red lines are the modeled change in DO as a result of the increase in temperature affecting both solubility and biological oxygen production/demand.**








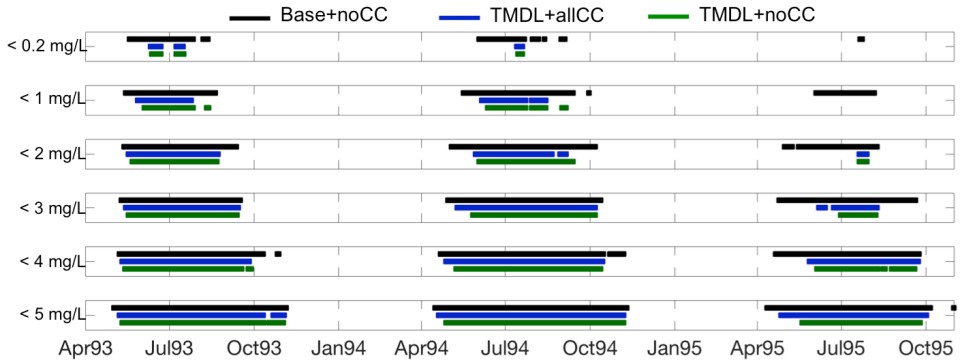

**Figure 7: Bars showing duration of hypoxic volume ( > 1km³) at each DO level for the Base+noCC run and the TMDL+noCC and TMDL+allCC nutrient scenarios.**











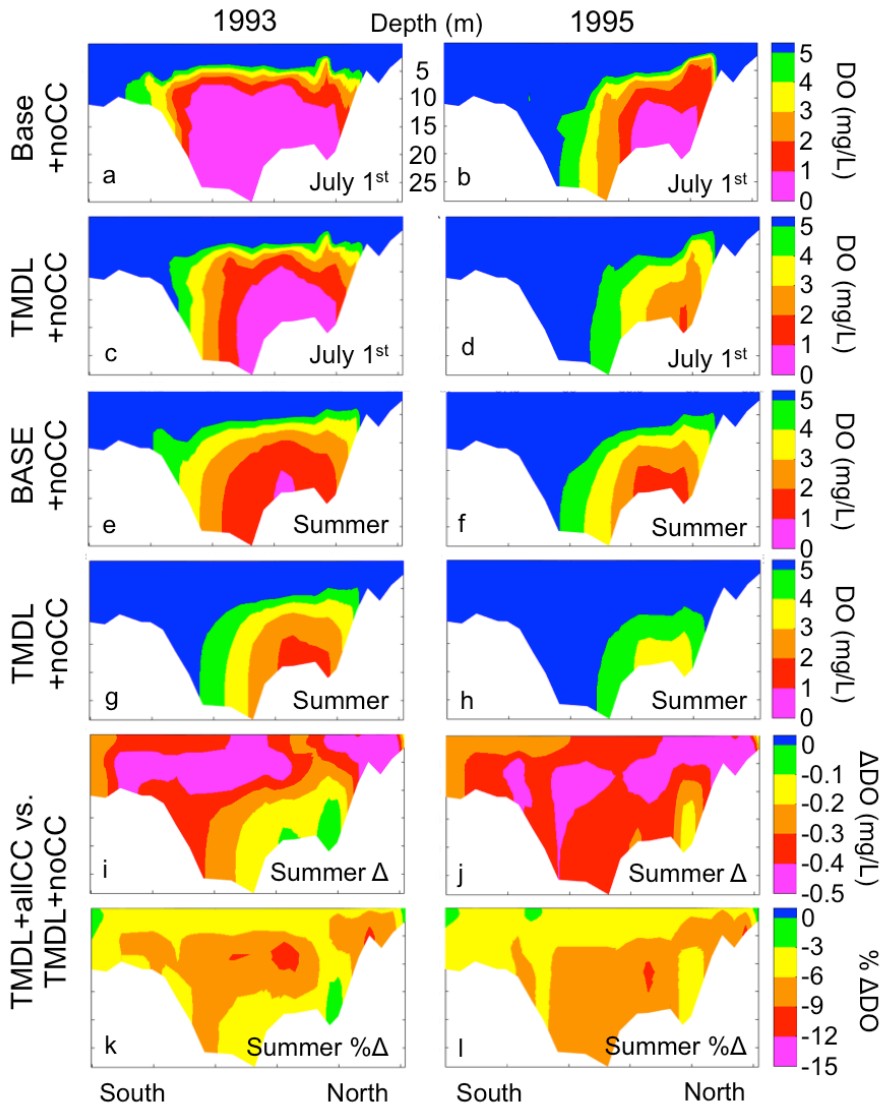

**Figure 8: Latitudinal along-Bay DO transects for the Base+noCC run and TMDL+noCC scenario for July 1, 1993 (a,c) and July 1, 1995 (b,d), average summer (May-Sept) for 1993 (e,g) and 1995 (f,h), the difference in average summer DO between the TMDL+noCC and TMDL+allCC scenarios (i,j), and the percent difference in average summer DO between the TMDL+noCC and TMDL+allCC scenarios (k,l).**





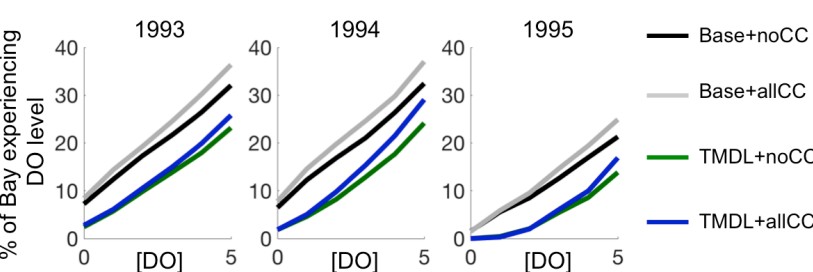


**Figure 9: Percent of the entire Bay that experiences a given DO level during 1993, 1994, and 1995.**











**Appendix A**

955   Before being used for climate change sensitivity experiments, the ChesROMS-ECB temperature
parameterizations were re-examined and modified as necessary based on information from the literature
and extensive skill assessment using data from 23 Chesapeake Bay Program Water Quality Monitoring
stations (Table A1). (Data are available at:
http://www.chesapeakebay.net/data/downloads/cbpwaterqualitydatabase1984present.) Modifications of

960 biological functions from the model version published in Feng et al. (2015) and used in the model
comparisons published in Irby et al. (2016) are documented in Table A2. Specifically, temperature
dependence was added to the zooplankton maximum growth rate, the remineralization rates of large and
small detritus, and the phytoplankton growth rate at temperatures above 20°C. The maximum rate of
nitrification, the temperature dependency on remineralization of semi-labile DON, and the remineralization

965 rate of DOC at 0°C were also modified to fit with current understanding (Lomas et al., 2002).

   Skill of the modified model was assessed via total Root Mean Squared Difference (RMSD; Table
A3), normalized target diagrams (Joliff et al., 2009), and time series analysis (Irby, 2017). For the total
RMSD calculations, the model results were compared to monthly/bi-monthly observations at the stations
and regions shown in Fig. 1. Results from the modified model were also compared to an earlier iteration of

970 the model evaluated in Irby et al. (2016).




**Table A1** Characteristics of observation stations.

| Station | Latitude (°N) | Longitude (°W) | Station Depth (m) | Region |
|---------|---------|----------|-------|--------|
| CB1.1 | 39.54794 | -76.08481 | 6.1 | A |
| CB2.1 | 39.44149 | -76.02599 | 6.3 | A |
| CB2.2 | 39.34873 | -76.17579 | 12.4 | A |
| CB3.1 | 39.2495 | -76.2405 | 13 | A |
| CB3.2 | 39.16369 | -76.30631 | 12.1 | B |
| CB3.3C | 38.99596 | -76.35967 | 24.3 | B |
| CB4.1C | 38.82593 | -76.39945 | 32.2 | B |
| CB4.2C | 38.64618 | -76.42127 | 27.2 | B |
| CB4.3C | 38.55505 | -76.42794 | 26.9 | B |
| CB4.4 | 38.41457 | -76.34565 | 30.3 | B |
| CB5.1 | 38.3187 | -76.29215 | 34.1 | C |
| CB5.2 | 38.13705 | -76.22787 | 30.6 | C |
| CB5.3 | 37.91011 | -76.17137 | 26.9 | C |
| CB5.4 | 37.80013 | -76.17466 | 31.1 | C |
| CB5.5 | 37.6918 | -76.18967 | 17 | C |
| CB6.1 | 37.58847 | -76.16216 | 12.5 | D |
| CB6.2 | 37.4868 | -76.15633 | 10.5 | D |
| CB6.3 | 37.41153 | -76.15966 | 11.3 | D |
| CB6.4 | 37.23653 | -76.20799 | 10.2 | D |
| CB7.1 | 37.68346 | -75.98966 | 20.9 | D |
| CB7.2 | 37.41153 | -76.07966 | 20.2 | D |
| CB7.3 | 37.11681 | -76.12521 | 13.6 | D |
| CB7.4 | 36.9957 | -76.02048 | 14.2 | D |





**Table A2**

| Symbol | Description | Feng et al. (2015) | Chapter 4 | Units |
|---|---|---|---|---|
| $g_{max}$ | *Zooplankton maximum growth rate | 0.3 | $0.05*e^{0.0742*T}$ | d$^{-1}$ |
| $n_{max}$ | Maximum rate of nitrification | 0.05 | 0.2 | d$^{-1}$ |
| $r_{D_L}$ | *Remineralization of large nitrogen detritus | 0.2 | $0.05*e^{0.0742*T}$ | d$^{-1}$ |
| $r_{D_S}$ | *Remineralization of small nitrogen detritus | 0.2 | $0.05*e^{0.0742*T}$ | d$^{-1}$ |
| $\kappa_{[DON]_{SL}}$ | *Temperature dependency remineralization of semi-labile DON | 0.07 | 0.0742 | (°C)$^{-1}$ |
| $a_{0C}$ | Remineralization rate of DOC at 0 °C | 0.003835 | 0.008 | d$^{-1}$ |
| $\mu_0$ | ^Phytoplankton growth rate | 2.15 | <20°C, 2.15 <br> T ≥20°C, $1.81 + e^{0.16*T-4.27}$ | d$^{-1}$ |

*Community respiration and zooplankton grazing temperature dependent functions are based on a Q$_{10}$ of 2.1 (Lomas et al., 2002)

^Phytoplankton growth rate at low temperatures (T < 20°C) is constant with higher temperatures following a rate based on Lomas et al. (2002) with a Q$_{10}$ from 20°C to 40°C of 2.18.













**Table A3** Total RMSD (and observational mean) of surface and bottom temperature (T), salinity (S), dissolved oxygen (DO) and nitrate ($NO_3$) of the present model and the earlier model version used in Feng et al. (2015) and Irby et al. (2016).

| Variable | Earlier Model Version | | | | | | Modified Model Version used here | | | | | |
| --- | --- | --- | --- | --- | --- | --- | --- | --- | --- | --- | --- | --- |
| | Total | A | B | C | D | | Total | A | B | C | D | |
| Surface T ($^\circ$C) | 1.23 (17.15) | 1.69 (16.86) | 0.91 (16.71) | 1.17 (17.25) | 1.15 (17.62) | | 1.23 (17.15) | 1.73 (16.86) | 0.92 (16.71) | 1.19 (17.25) | 1.17 (17.62) | |
| Bottom T ($^\circ$C) | 2.27 (15.98) | 1.68 (16.74) | 2.92 (15.42) | 2.28 (15.89) | 1.88 (16.09) | | 2.22 (15.98) | 1.72 (16.74) | 2.84 (15.42) | 2.22 (15.89) | 1.86 (16.09) | |
| Surface S | 2.14 (12.64) | 2.32 (1.84) | 1.96 (10.54) | 1.95 (14.20) | 2.30 (19.15) | | 1.86 (12.64) | 2.11 (1.84) | 1.56 (10.54) | 1.54 (14.20) | 2.11 (19.15) | |
| Bottom S | 2.09 (18.18) | 2.10 (3.92) | 1.78 (17.59) | 2.04 (20.88) | 2.34 (24.52) | | 2.17 (18.18) | 2.31 (3.92) | 2.04 (17.59) | 1.83 (20.88) | 2.40 (24.52) | |
| Surface DO (mg L$^{-1}$) | 1.60 (9.35) | 1.89 (9.24) | 1.61 (9.65) | 1.56 (9.39) | 1.43 (9.13) | | 1.48 (9.35) | 1.73 (9.24) | 1.61 (9.65) | 1.28 (9.39) | 1.34 (9.13) | |
| Bottom DO (mg L$^{-1}$) | 2.51 (5.78) | 2.62 (8.00) | 1.63 (3.82) | 2.49 (5.01) | 2.95 (6.71) | | 1.82 (5.78) | 2.47 (8.00) | 1.63 (3.82) | 1.58 (5.01) | 1.72 (6.71) | |
| Surface $NO_3$ (mmolN m$^{-3}$) | 0.21 (0.32) | 0.36 (0.93) | 0.18 (0.34) | 0.17 (0.17) | 0.13 (0.07) | | 0.20 (0.32) | 0.35 (0.93) | 0.18 (0.34) | 0.13 (0.17) | 0.11 (0.07) | |
| Bottom $NO_3$ (mmolN m$^{-3}$) | 0.17 (0.21) | 0.34 (0.81) | 0.28 (0.14) | 0.07 (0.08) | 0.07 (0.04) | | 0.23 (0.21) | 0.37 (0.81) | 0.28 (0.14) | 0.14 (0.08) | 0.06 (0.04) | |



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

A Report to the US Environmental Protection Agency Chesapeake Bay Program and
    to The US Army Engineer Baltimore District, US Army Engineer Research and
    Development Center, Vicksburg, MS, 2010.

Chua, V.P. and Xu, M.: Impacts of sea-level rise on estuarine circulation: An idealized
    estuary and San Francisco Bay, J. Mar. Sys., 139: 58-67, doi:
10.1016/j.jmarsys.2014.05.012, 2014.

Church, J.A., P.U. Clark, A. Cazenave, J.M. Gregory, S. Jevrejeva, A. Levermann, M.A.
    Merrifeld, G.A. Milne, R.S. Nerem, P.D. Nunn, A.J. Payne, W.T. Pfeffer, D.
    Stammer, and A.S. Unnikrishnan, 2013: Sea Level Change. In: *Climate Change*
    *2013: The Physical Science Basis. Contribution of Working Group I to the Fifth*



1090  *Assessment Report of the Intergovernmental Panel on Climate Change*, Cambridge
    University Press.

    Collins, M., Knutti, R., Arblaster, J., Dufresne, J-L., Fichefet, T., Friedlingstein, P., Gao,
    X., Gutowski, W. J., Johns, T., Krinner, G., Shongwe, M., Tebaldi, C., Weaver, A. J.,
    and Wehner, M.: Long-term Climate Change: Projections, Commitments and
1095  Irreversibility, Climate Change 2013: The Physical Science Basis, Contribution of
    Working Group I to the Fifth Assessment Report of the Intergovernmental Panel on
    Climate Change, Cambridge University Press, Cambridge, United Kingdom and New
    York, NY, USA, 2013.

    Diaz, R. J. and Rosenberg, R.: Spreading dead zones and consequences for marine
1100  ecosystems, Science, 321, 926–929, doi:10.1126/science.1156401, 2008.

    Ding, H., and Elmore, A.J.: Spatio-temporal patterns in water surface temperature from
    Landsat time series data in the Chesapeake Bay, U.S.A., Remote Sens. Environ., 168:
    335-348. doi: 10.1016/j.rse.2015.07.009, 2015.

    Feng, Y., Friedrichs, M. A. M., Wilkin, J., Tian, H., Yang, Q., Hofmann, E. E., Wiggert,
1105  J. D., and Hood, R. R.: Chesapeake Bay nitrogen fluxes derived from a land-estuarine
    ocean biogeochemical modeling system: model description, evaluation, and nitrogen
    budgets, J. Geophys. Res.-Biogeo., 120, 1666–1695, doi:10.1002/2015JG002931,
    2015.

    Goberville, E., G. Beaugrand, N-C. Hautekeete, Y. Piquot, and C. Luczak.: Uncertainties
1110  in the projection of species distributions related to general circulation models, Ecol.
    Evol., 5: 1100-1116. doi: 10.1002/ece3.1411, 2015.

    Groisman, P.Y., Knight, R.W., Karl., T.R., Easterling, D.R., Sun, B., and Lawrimore,
    J.H.: Contemporary changes of the hydrological cycle over the contiguous United
    States: Trends derived from in situ observations, J. Hydeometeorol., 5, 1, 64-85,
1115  2004.

    Hagy, J. D., Boyton, W. R., Keefe, C. W., and Wood, K. V.: Hypoxia in Chesapeake
    Bay, 1950–2001: long-term change in relation to nutrient loading and river flow,
    Estuaries, 27, 634–658, 2004.

    Harding Jr., L. W., Gallegos, C. L., Perry, E. S., Miller, W. D., Adolf, J. E., Mallonee, M.
1120  E., and Paerl, H. W.: Long-term trends of nutrients and phytoplankton in Chesapeake



Bay, Estuar. Coast., doi:10.1007/s12237-015-0023-7, online first, 2015.

Harding, L.W., Gallegos, C.L., Perry, E.S., Miller, W.D., Adolf, J.E., Mallonee, M.E., and Paerl, H.W.: Long-Term Trends of Nutrients and Phytoplankton in Chesapeake Bay, Estuar. Coast., 39, 664-681, doi:10.1007/s12237-015-0023-7, 2016.

Hargreaves, G.H, and Samani, Z.A.: Estimating potential evapotranspiration, Journal of the Irrigation and Drainage Division, 108: 225-230, 1982.

Hofmann, E.E., B. Cahill, K. Fennel, M.A.M. Friedrichs, K. Hyde, C. Lee, A. Mannino, R.G. Najjar, J.E. O'Reilly, J. Wilkin, and J. Xue.: Modeling the Dynamics of Continental Shelf Carbon, Annu. Rev. Mar. Sci., 3: 93-122. doi: 10.1146/annurev-
marine-120709-142740, 2011.

Hong, B. and Shen, J.: Responses of estuarine salinity and transport processes to potential future sea-level rise in the Chesapeake Bay, Estuar. Coast. Shelf S., 104–105, 33–45, doi:10.1016/j.ecss.2012.03.014, 2012.

IPCC, 2013: Annex 1: Atlas of Global and Regional Climate Projections [van
Oldenborgh, G.J., M. Collins, J. Arblaster, J.H. Christensen, J. Marotzke, S.B. Power, M. Rummukainen, and T. Zhou (eds.]. In: Climate Change 2013: The Physical Science Basis. Controbutions of the Working Group 1 to the Fifth Assessment Report of the Intergovernmental Panel on Climate Change [Stocker, T.F., D. Qin, G-K. Plattner, M. Tignor, S.K. Allen, J.Boschung, A. Nauels, Y. Xia, V. Bex, and P.M.
Midgley (eds.)]. Cambridge University Press, Cambridge, United Kingdom and New York, NY, USA, 2013.

IPCC, 2013: Summary for Policymakers, In: Climate Change 2013: The Physical Science Basis. Contribution of Working Group I to the Fifth Assessment Report of the Intergovernmental Panel on Climate Change [Stocker, T.F., D. Qin, G-K. Plattner, M.
Tignor, S.K. Allen, J. Boschung, A. Nauels, Y. Xia, V. Bex, and P.M. Midgley (eds.)]. Cambridge University Press, Cambridge, United Kingdom and New York, NY, USA, 2013.

Irby, I.D., Friedrichs, M.A.M., Friedrichs, C.T., Bever, A.J., Hood, R.R., Lanerolle, L.W.J., Li, M., Linker, L., Scully, M.E., Sellner, K., Shen, J., Testa, J., Wang, H.,
Wang, P. and Xia, M.: Challenges associated with modeling low-oxygen waters in Chesapeake Bay: a multiple model comparison, Biogeosciences, 13, 2011-2018,



doi:10.5194/bg-13-2011-2016, 2016.

Irby, I.D. and Friedrichs, M.A.M.: Evaluating confidence in the impact of regulatory
nutrient reduction on Chesapeake Bay water quality, Estuar. Coast., in revision.

Irby, I.D.: Using Water Quality Models in Management – A Multiple Model Assessment,
Analysis of Confidence, and Evaluation of Climate Change Impacts, Doctoral thesis,
William & Mary, Williamsburg, VA, USA, 2017.

Justic, D., Bierman, V.J., Scavia, D., and Hetland, R.D.: Forecasting Gulf's Hypoxia: The
Next 50 Years?, Estuar. Coast., 30, 5, 791-801, 2007.

Keisman, J. and Shenk, G.: Total maximum daily load criteria assessment using
monitoring and modeling data, J. Am. Water Re- sour. As., 49, 1134–1149,
doi:10.1111/jawr.12111, 2013.

Kemp, W. M., Testa, J. M., Conley, D. J., Gilbert, D., and Hagy, J. D.: Temporal
responses of coastal hypoxia to nutrient loading and physical controls,

Biogeosciences, 6, 2985–3008, doi:10.5194/bg-6-2985-2009, 2009.

Lake, S.J. and Brush, M.J.: Modeling estuarine response to load reductions in a warmer
climate: York River Estuary, Virginia, USA, Mar. Ecol. Prog. Ser., 538: 81-98. doi:
10.3354/meps11448, 2015.

Lapointe, D., W.K. Vogelbein, M.C. Fabrizio, D.T. Gauthier, and R.W. Brill.:

Temperature, hypoxia, and mycobacteriosis: effects on adult striped bass *Morone
saxatilis* metabolic performance, Dis. Aquat. Organ., 108: 113-127. doi:
10.3354/dao02693, 2014.

Lee, M., E. Shevliakova, S. Malyshev, P.C.D. Milly, and P.R. Jaffe., Climate variability
and extremes, interacting with nitrogen storage, amplify eutrophication risk,

Geophys. Res. Lett., 43: 7520-7528. doi: 10.1002/2016GL069254, 2016.

Lomas, M.W., P.M. Gilbert, F.K. Shiah, and E.M. Smith.: Microbial processes and
temperature in Chesapeake Bay: current relationships and potential impacts of
regional warming, Glo. Change Biol., 8: 51-70, 2002.

McCormick, S., Simmens, S.J., Glicksman, R.L., Paddock, L., Kim, D., Whited, B., and

Davies, W.: Science in litigation, the third branch of U.S. climate policy, Science,
357, 6355, 979-980, doi:10.1126/science.aao0412, 2017.



Meier, H.E.M., H.C. Anderson, K. Eilola, B.G. Gustafsson, I. Kuznetsov, B. Muller-Karulis, T. Neumann, and O.P. Savchuk.: Hypoxia in future climates: A model ensemble study for the Baltic Sea, Geophys. Res. Lett., 38. doi: 10.1029/2011GL049929, 2011.

Meire, L., Soetaert, K. E. R., and Meysman, F. J. R.: Impact of global change on coastal oxygen dynamics and risk of hypoxia, Biogeosciences, 10, 2633–2653, doi:10.5194/bg-10-2633-2013, 2013.

Muhling, B.A., C.F. Gaitan, C.A. Stock, V.S. Saba, D. Tommasi, and K.W. Dixon.: Potential salinity and temperature futures for the Chesapeake Bay using a statistical downscaling spatial disaggregation framework, Estuar. Coast., doi: 10.1007/s12237-017-0280-8, 2017.

Murphy, R. R., Kemp, W. M., and Ball, W. P.: Long-term trends in Chesapeake Bay seasonal hypoxia, stratification, and nutrient loading, Estuar. Coast., 34, 1293–1309, doi:10.1007/s12237- 011-9413-7, 2011.

Najjar, R.G., L. Patterson, and S. Graham.: Climate Simulations of Major Estuarine Watersheds in the Mid-Atlantic Region of the US, Climate Change, 95, 2009.

Najjar, R. G., Pyke, C. R., Adams, M. B., Breitburg, D., Hersh- ner, C., Kemp, M., Howarth, R., Mulholland, M. R., Paolisso, M., Secor, D., Sellner, K., Wardrop, D., and Wood, R.: Potential climate-change impacts on the Chesapeake Bay, Estuar. Coast. Shelf S., 86, 1–20, doi:10.1016/j.ecss.2009.09.026, 2010.

Portner, H.O. and Knust, R.: Climate change affects marine fishes through the oxygen limitation of thermal tolerance, Science, 315: 95-97, 2007.

Portner, H.O. and Lanning, G.: Oxygen and capacity limited thermal tolerance, The hypoxic environment, Fish Physiol., Vol 27, Academic Press, San Diego, CA, p 143-191, 2009.

Preston, B.L.: Observed Winter Warming of the Chesapeake Bay Estuary (1949-2002): Implications for Ecosystem Management, Environmental Assessment, 34: 125-139. doi: 10.1007/s00267-004-0159-x, 2004.

Reclamantion: Downscaled CMIP3 and CMIP5 Climate and Hydrology Projections: Release of Downscaled CMIP5 Climate Projections, Comparison with preceding Information, and Summary of User Needs. Prepared by the U.S. Department of the



Interior, Bureau of Reclamation, Technical Services Center, Denver, Colorado. 47pp, 2013.

Ross, A.C., R.G. Najjar, M. Li, M.E. Mann, S.E. Ford and B. Katz.: Sea-level rise and other influences on decadal-scale salinity variability in a coastal plain estuary, Estuar. Coast. Shelf S., 157: 79-92. doi: 10.1016/j.ecss.2015.01.002, 2015.

Saba, V.S., Griffies, S.M., Anderson, W.G., Winton, M., Alexander, M.A., Delworth, T.L., Hare, J.A., Harrison, M.J., Rosati, A., Vecchi, G.A., and Zhang, R.: Enhanced

warming of the Northwest Atlantic Ocean under climate change, J. Geophys. Res.-Oceans, 120, doi:10.1002/2015JC011346, 2015.

Sallenger, A.H., K.S. Doran, and P.A. Howd.: Hotspot of accelerated sea-level rise on the Atlantic coast of North America, Nat. Clim. Change, doi: 10.1038/NCLIMATE1597, 2012.

Scully, M. E.: The importance of climate variability to wind-driven modulation of hypoxia in Chesapeake Bay, J. Phys. Oceanogr., 40, 1435–1440, doi:10.1175/2010JPO4321.1, 2010.

Shchepetkin, A. F. and McWilliams, J. C.: The Regional Ocean Modeling System (ROMS): a split-explicit, free-surface, topography-following-coordinate oceanic

model, Ocean Model., 9, 347–404, doi:10.1016/j.ocemod.2004.08.002, 2005.

Shenk, G. W. and Linker, L. C.: Development and application of the 2010 Chesapeake Bay watershed total maxi- mum daily load model, J. Am. Water Resour. As., 49, 1–15, doi:10.1111/jawr.12109, 2013.

Sinha, E., A.M. Michalak, and V. Balaji.: Eutrophication will increase during the 21st

century as a result of precipitation changes, Science, 357: 405-408. doi: 10.1126/science.aan2409, 2017.

Son, S.H. and Wang, M.: Diffuse attenuation coefficient of the photosynthetically available radiation $K_d$(PAR) for global open ocean and coastal waters, Rem. Sen. Environ., 159: 250-258. doi: 10.1016/j.rse.2014.12.011, 2015.

Sweet, W.V., R.E. Kopp, C.P. Weaver, J. Obeysekera, R.M. Horton, E.R. Thieler, and C. Zervas.: Global and Regional Sea Level Rise Scenarios for the United States. NOAA Technical Report NOS CO-OPS 083, 2017.





Tango, P.J. and Batiuk, R.A.: Deriving Chesapeake Bay water quality standards, J. Am.
Water Resour. As., 1-18. doi: 10.1111/jawr.12108, 2013.

Tian, H., Yang, Q., Najjar, R., Ren, W., Friedrichs, M. A. M., Hop- kinson, C. S., and
Pan, S.: Anthropogenic and climatic influences on carbon fluxes from eastern North
America to the Atlantic Ocean: a process-based modeling study, J. Geophys. Res.-
Biogeo., 120, 752–772, doi:10.1002/2014JG002760, 2015.

USEPA: Chesapeake Bay Total Maximum Daily Load for Nitrogen, Phosphorus, and
Sediment, US Environmental Protection Agency, US Environmental Protection
Agency Chesapeake Bay Program Office, Annapolis, MD, 2010.

Vaquer-Sunyer, R. and Duarte, C. M.: Thresholds of hypoxia for marine biodiversity, P.
Natl. Acad. Sci. USA, 105, 15452–15457, doi:10.1073/pnas.0803833105, 2008.

Vaquer-Sunyer, R. and Duarte, C.M.: Temperature effects on oxygen thresholds for
hypoxia in marine benthic organisms, Glob. Change Biol., doi: 10.1111/j.1365-
2486.2010.02343.x, 2011.

Wang, M., S.H. Son, and L.W. Harding.: Retrieval of diffuse attenuation coefficient in
the Chesapeake Bay and turbid ocean regions for satellite ocean color applications, J.
Geophys. Res.– Oceans, 114. doi: 10.1029/2009JC005286, 2009.

Winder, M. and Sommer, U.: Phytoplankton response to a changing climate,
Hydrobiologia, 698: 5-16, 2012.

Xu, J., Long, W., Wiggert, J. D., Lanerolle, L. W. J., Brown, C. W., Murtugudde, R., and
Hood, R. R.: Climate forcing and salinity variability in Chesapeake Bay, USA,
Estuar. Coast. Shelf S., 35, 237–261, doi:10.1007/s12237-011-9423-5, 2012.

Yang, Q., Tian, H., Friedrichs, M. A. M., Hopkinson, C. S., Lu, C., and Najjar, R. G.:
Increased nitrogen export from eastern North America to the Atlantic Ocean due to
climatic and anthro- pogenic changes during 1901–2008, J. Geophys. Res.-Biogeo.,
120, 1046–1068, doi:10.1002/2014JG002763, 2015a.

Yang, Q., Tian, H., Friedrichs, M. A. M., Liu, M., Li, X., and Yang, J.: Hydrological
responses to climate and land-use changes along the North American east coast: a
110-year his- torical reconstruction, J. Am. Water Resour. As., 51, 47–67,
doi:10.1111/jawr.12232, 2015b.