# Peer review of "The competing impacts of climate change and nutrient reductions on dissolved oxygen in Chesapeake Bay"

_Biogeosciences, 2017_

## Referee Comment (RC1) · Anonymous Referee #1 · 5 Nov 2017

This manuscript reports on the effect of reduced nutrient loads and climate change scenarios on the degree of hypoxia in Chesapeake Bay using watershed and estuarine models. Such studies can rapidly blow out in complexity and become very hard to communicate concisely. However, the authors skilfully undertook a very nice study that that was well written and structured and broke down some of the likely effects of climate change on hypoxia and compared this to improvements from nutrient load reductions. I found these questions very pertinent based on my recent interactions with coastal management authorities and I think this manuscript will probably a very well cited one in the field.

The model has already been published and was slightly modified for the purposes of this study. I think it was adequately referenced and described.

It was particularly interesting to note that sea level rise could lead to a slight increase in ventilation, and hence increase in DO. As expected, however the overwhelming effect was the reduction in O2 solubility. I also found it interesting and important that the nutrient reductions proved to be worthwhile in the face of climate change (this is a novel and important part of the study). Such modelling exercises have a high degree of uncertainty, but I think this was well addressed in the discussion. The figures were well presented and struck a good balance between clarity and the overwhelming amount of data that such a study invariably produces. It was a pleasure to read such a well written and organised manuscript on a complex topic. I have only some very minor editorial comments.

A little more information on the amount of nutrient reduction occurring for the TMDL scenarios would be useful. % reduction is fine.

Line 183 effects Line 335, I would not say the decrease in hypoxic duration was large except in a few specific instances. Perhaps moderate would be a better word?

Line 483 'made a first order assessment' might sound a bit less casual than 'took a first order look'

Table A2, no caption, chapter 4 needs to be defined here.

Table A3, Total, A,B,C and D need to be defined in the caption

---

## Referee Comment (RC2) · Anonymous Referee #2 · 14 Nov 2017

The study on impacts of climate change and nutrient reductions on dissolved oxygen in Chesapeake Bay is well written and addresses an interesting topic. However, there are many shortcomings of the chosen approach:

1) The simulations did not consider the impact of temperature changes on hydrodynamics. Wind and evaporation did not change be definition. Only the impacts of increases in air temperature, global sea level, river flow and nutrient loads (related to river flow) were considered. Hence, the approach is not dynamically consistent.

2) A time slice approach was chosen and the transient behavior was neglected. A period of only three years was investigated. Uncertainty caused by natural variability

was not investigated. In particular, the impacts of the large variability of sea level pressure and wind fields on the simulation results were not considered because the time slices are too short. Hence, it is not clear to me whether the calculated changes in dissolved oxygen concentrations are statistically significant.

3) The applied changes in air temperature, sea level and river flow were estimated from ensemble mean values from global model simulations from the literature (partly grey literature) and are not consistent results of changing climate in the region. In particular, it was impossible for me to understand how the watershed simulations were done. A regional climate model with sufficient horizontal resolution was not applied. Hence, the simulations are not dynamically consistent projections. I suggest to call them sensitivity studies.

4) The uncertainty of projected future climate caused by biases of global climate models was not assessed. Usually there is a large spread of projected changes around the ensemble mean. The spread of the calculated changes in dissolved oxygen concentrations may be larger than the differences between the impact of nutrient load changes and the impact of climate change on the results. Hence, the conclusions on the competing impacts may not hold in a multi-model ensemble approach.

5) A greenhouse gas emission scenario RCP 4.5 was chosen. The question whether the conclusions would also hold for RCP 8.5, which is not necessarily less likely than RCP 4.5, was not addressed.

6) Quantitative figures of changing nutrient concentrations and nutrient supply are not given. Hence, it is difficult to compare with other coastal seas with comparable environmental situation.

7) Why has sea level rise a positive impact on dissolved oxygen concentrations in regions B and C? This result is unexpected. The authors state that the impact of increased stratification and residence time is smaller than the impact of increased estuarine circulation. Is this result supported by other model studies or possibly a shortcoming of the present model?

In summary, the present study is not about changing climate with all its uncertainties and the approach does not support the provided conclusions that otherwise would have large impact on marine management. Hence, I recommend to perform longer simulations to estimate uncertainties caused by natural variability (usually 30-year long simulations are recommended to address the statistics of weather). To estimate uncertainties due to model biases an ensemble of simulations driven by various global model results should be performed. Further, a high-end emission scenario like RCP 8.5 should be investigated to be able to conclude (perhaps) that the impact of changing climate does not counteract the impact of nutrient load reductions. I recommend a major revision.

---

## Short Comment (SC1) · 16 Nov 2017

We would like to thank Anonymous Reviewer #1 for their positive assessment of our manuscript. We agree that it was a challenge to communicate these complex results concisely; this was certainly our goal, and we are pleased to hear that the Reviewer thought we succeeded in this regard. We also appreciate the minor editorial comments provided, which clearly demonstrate the reviewer carefully read the manuscript (even the appendix!) These comments will be addressed after the open discussion period concludes and will improve the manuscript for future readers.

---

## Short Comment (SC2) · 18 Nov 2017

Reply to Reviewer #2:

We would like to thank Anonymous Reviewer #2 for their review of our manuscript, and are glad to hear that they found the manuscript to be well written. As Reviewer #1 noted, it is a challenge to present such complex results (with so many variables changing in time and x,y,z space) in a concise manner.

The reviewer's comments, however, make us feel that perhaps the overall objectives of our study may not have been clear. Our study is structured as an initial exploration of the potential ramifications of the *first order* impacts of climate change on oxygen concentrations in the Chesapeake Bay. In the Chesapeake Bay region many researchers, ourselves included, are working on long-term ~50-year simulations that include all possible climate effects: changes in solar radiation, humidity, and winds for example, in addition to the other effects examined here (changes in temperature, precipitation and sea level rise). This initial study, however, opts for a different and less complex approach, whereby the impacts of a few first order factors are studied in detail via sensitivity analysis. We by no means have meant to imply that we are predicting what the Bay will look like in 2050. We leave this to future work that is focusing on incorporating all climate change effects simultaneously, and running realistic 50+year simulations.

In the first sentence of our "*Methodological Limitations*" section we explicitly state: "*This research is a first order look at the potential impacts that changes in climate may have on the efficacy of nutrient reduction efforts in the Chesapeake Bay; however, more robust examinations of the problem are needed in order to adequately aid in the regulatory decision making process going forward.*" We also end this section by saying: "*To address these limitations, an effort to conduct a continuous simulation from 2015 – 2050 including both gradual implementation of the nutrient reductions and climate change impacts is currently underway.*" However the reviewer's comments indicate to us that this clearly needs to be explained up front in the introduction as well. Thus, for the next version of our manuscript we are making modifications to our introduction to make our objectives for this analysis, and our future work, clearer for future readers.

We appreciate the opportunity to address each of the reviewer's specific comments below.

*1) The simulations did not consider the impact of temperature changes on hydrodynamics. Wind and evaporation did not change be definition. Only the impacts of increases in air temperature, global sea level, river flow and nutrient loads (related to river flow) were considered. Hence, the approach is not dynamically consistent.*

As described on line 180, air temperature was actually not changed in our sensitivity studies. Instead, water temperature was changed consistently throughout the water column. This choice is rationalized in the following paragraph (lines 180-195) where we cite prior studies that have documented that surface and bottom waters of the Bay are warming uniformly and thus have limited impact on Bay hydrodynamics. Stratification in

the Chesapeake Bay is primarily governed by salinity, not temperature, and therefore future warming is not likely to significantly impact stratification. The significant impact of future temperatures on continental shelf and open ocean stratification is well known; however previous studies have indicated that this will likely not be a significant effect in Chesapeake Bay. Again, here we are only looking at first-order climate change impacts on DO – the impact of changes in temperature on solubility and growth/grazing/remineralization dynamics. Future work will look at the second-order effect of warming-induced hydrodynamic changes on hypoxia.

*2) A time slice approach was chosen and the transient behavior was neglected. A period of only three years was investigated. Uncertainty caused by natural variability was not investigated. In particular, the impacts of the large variability of sea level pressure and wind fields on the simulation results were not considered because the time slices are too short. Hence, it is not clear to me whether the calculated changes in dissolved oxygen concentrations are statistically significant.*

A "time slice" approach typically involves running a simulation at a future time interval, for example 2046-2050, rather than for a complete long-term simulation, e.g. 1990-2050. Here we opt for neither of these approaches, but instead adopt a third approach that involves looking at the sensitivity of a simulation to environmental changes. Our four-year simulations (one year spin-up, three year simulation) are not meant to be representative of 2046-2050. Instead we hold winds, humidity and solar radiation constant in order to look at the sensitivity to first-order environmental impacts. We must make this point clearer in our methods section, so that future readers are not confused about this point.

Also we note that natural interannual variability in Chesapeake Bay hypoxia is overwhelmingly dominated to first order by whether a particular year is characterized by higher than average rainfall (a "wet" year) or lower than average rainfall (a "dry" year). Here we carefully investigate both very wet and very dry years. We do document differences in results for the two "types" of years (recall our finding that a wet year with the TMDL nutrient reductions has more hypoxia than a dry year without the nutrient reductions), but generally our primary conclusions hold regardless of whether a year is particularly wet or dry.

*3) The applied changes in air temperature, sea level and river flow were estimated from ensemble mean values from global model simulations from the literature (partly grey literature) and are not consistent results of changing climate in the region. In particular, it was impossible for me to understand how the watershed simulations were done. A regional climate model with sufficient horizontal resolution was not applied. Hence, the simulations are not dynamically consistent projections. I suggest to call them sensitivity studies.*

Yes, we absolutely agree that throughout the manuscript we are performing "sensitivity studies" and are not providing "projections" for 2050. This is an important change that we must make throughout our manuscript, specifically in our introduction where we

describe the paper's objectives. We appreciate the reviewer pointing out this source of confusion.

We also apologize that our methods for deriving future river flow were not clear. We see the reviewer's confusion and feel that in our revised manuscript it will be better to state up front what changes in flow we are imposing (Table 2) and then describe how these estimates are consistent with what we know from the literature. This is analogous with what we did in the previous sections (2.3.1 and 2.3.2). This will also help make it clearer to future readers that we are indeed performing sensitivity experiments, and not trying to project what dissolved oxygen concentrations will actually be in 2050.

*4) The uncertainty of projected future climate caused by biases of global climate models was not assessed. Usually there is a large spread of projected changes around the ensemble mean. The spread of the calculated changes in dissolved oxygen concentrations may be larger than the differences between the impact of nutrient load changes and the impact of climate change on the results. Hence, the conclusions on the competing impacts may not hold in a multi-model ensemble approach.*

It is true that we did not assess the impacts of biases in the Global Climate Models, but this is because we are performing a sensitivity study, examining how sensitive Chesapeake Bay oxygen concentrations are to changes in water temperature, sea level rise and river flow.

Again, the goal of this paper, which absolutely needs to be clarified in our introductory paragraphs, is to provide a first look at the sensitivity of oxygen concentrations and hypoxic volume to these environmental forcing changes. Ideally the sensitivity of the estuary would be tested for a number of different temperature changes that would encompass uncertainties in future temperatures estimated by various GCMs and RCP scenarios (perhaps looking at a change of 1°C and 3°C as well as our 1.75°C experiment). An additional analysis that involves examining the increase in temperature required to completely nullify all positive impacts of the TMDL reduction (for DO < 5 mg/L) could be conducted. (Recall from Figure 4 that an increase of 1.75°C results in a 40% reduction in the gains of the TMDL. A temperature of 3 or 4°C might result in a complete negation of all TMDL gains.) Because our results show hypoxia is not very sensitive to changes in SLR and changes in river inputs, it is far less critical to examine how sensitive hypoxia is to varying levels of SLR and river inputs.

*5) A greenhouse gas emission scenario RCP 4.5 was chosen. The question whether the conclusions would also hold for RCP 8.5, which is not necessarily less likely than RCP 4.5, was not addressed.*

As the reviewer notes, we report to only use the RCP4.5 scenario. However, in our revised manuscript we will reword this to say that we are performing sensitivity experiments that are generally representative of what might be expected in a RCP4.5 scenario. In fact the temperature change (1.75°C) we choose to examine is also generally consistent with what might be expected for RCP8.5, since, as we note in the

manuscript: "for 2050 projections, studies have demonstrated that the difference between RCP scenarios is smaller than the spread of individual global climate models that utilize the RCP emission scenarios (e.g., Goberville et al., 2015)." Thus, if we were examining conditions in 2100, it would be more important to examine multiple RCP scenarios than multiple Climate Models, but in 2050 it is more important to examine multiple Climate models than multiple RCP scenarios. The next version of our manuscript will make it clearer that although our estimates of future change (required for our sensitivity studies) are broadly consistent with RCP4.5 assumptions, they are not very different from what we would expect for RCP8.5.

*6) Quantitative figures of changing nutrient concentrations and nutrient supply are not given. Hence, it is difficult to compare with other coastal seas with comparable environmental situation.*

This is an excellent point that reviewer 1 brought up as well. The revised manuscript will contain an additional table that includes the total amount of inorganic nitrogen entering the Bay for each experiment.

*7) Why has sea level rise a positive impact on dissolved oxygen concentrations in regions B and C? This result is unexpected. The authors state that the impact of increased stratification and residence time is smaller than the impact of increased estuarine circulation. Is this result supported by other model studies or possibly a short-coming of the present model?*

This was a surprising result to us as well, though we were reassured when presentations by the Chesapeake Bay Program's modeling team showed the same result for their preliminary Mid-Point Assessment simulations (e.g. https://www.chesapeakebay.net/channel_files/25275/purpose_of_wqstm_overview_6-5-17.pdf). We feel that we can perhaps do a better job of explaining this result in the manuscript. Overall, the increase in sea level at the model's open boundary (essentially a deeper opening at the Chesapeake Bay mouth) causes a significantly greater transport of oxygenated ocean water into the estuary at depth and a greater surface transport of water out of the estuary.

*In summary, the present study is not about changing climate with all its uncertainties and the approach does not support the provided conclusions that otherwise would have large impact on marine management. Hence, I recommend to perform longer simulations to estimate uncertainties caused by natural variability (usually 30-year long simulations are recommended to address the statistics of weather). To estimate uncertainties due to model biases an ensemble of simulations driven by various global model results should be performed. Further, a high-end emission scenario like RCP 8.5 should be investigated to be able to conclude (perhaps) that the impact of changing climate does not counteract the impact of nutrient load reductions. I recommend a major revision.*

We completely agree with the reviewer that the present study is not about changing

climate with all its uncertainties. As stated earlier, we must make this clearer to our readers up front in the introduction. Instead, it presents sensitivity studies which illustrate the impact that three first order environmental variables may have on Chesapeake Bay oxygen concentrations in the future: water temperature, sea level rise and river flow. As discussed in our "*Methodological Limitations*" section, this work is being followed by a larger study involving experts in global climate models and downscaling techniques. This future study will indeed involve longer-term simulations (1985-2050), address uncertainties in climate model biases, and directly include changes in humidity, solar radiation and winds, in addition to the variables investigated here. Nevertheless, we feel that our results here are robust and worthy of publication as they have clearly established several new results that have not been published before, namely that: (1) the potential impacts of climate change will be significantly smaller than improvements in DO expected in response to the required nutrient reductions, especially at the anoxic and hypoxic levels, and (2) increased temperature exhibits the strongest control on the change in future DO concentrations, while sea level rise is expected to exert a small positive impact and increased winter river flow is anticipated to exert a small negative impact.

Our revisions in response to Reviewer #2's comments will make this a much stronger manuscript; we are very appreciative of the time spent reviewing and providing these comments.

---

## Author Comment (AC1) · 23 Dec 2017

**Author responses to the Referee's comments are included in blue.**

**Anonymous Referee #1**

This manuscript reports on the effect of reduced nutrient loads and climate change scenarios on the degree of hypoxia in Chesapeake Bay using watershed and estuarine models. Such studies can rapidly blow out in complexity and become very hard to communicate concisely. However, the authors skilfully undertook a very nice study that that was well written and structured and broke down some of the likely effects of climate change on hypoxia and compared this to improvements from nutrient load reductions. I found these questions very pertinent based on my recent interactions with coastal management authorities and I think this manuscript will probably a very well cited one in the field.

The model has already been published and was slightly modified for the purposes of this study. I think it was adequately referenced and described.

It was particularly interesting to note that sea level rise could lead to a slight increase in ventilation, and hence increase in DO. As expected, however the overwhelming effect was the reduction in O2 solubility. I also found it interesting and important that the nutrient reductions proved to be worthwhile in the face of climate change (this is a novel and important part of the study). Such modelling exercises have a high degree of uncertainty, but I think this was well addressed in the discussion. The figures were well presented and struck a good balance between clarity and the overwhelming amount of data that such a study invariably produces. It was a pleasure to read such a well written and organised manuscript on a complex topic. I have only some very minor editorial comments.

We would like to thank Anonymous Reviewer #1 for their positive assessment of our manuscript. We agree that it was a challenge to communicate these complex results concisely; this was certainly our goal, and we are pleased to hear that the Reviewer thought we succeeded in this regard.

A little more information on the amount of nutrient reduction occurring for the TMDL scenarios would be useful. % reduction is fine.

Yes, we agree that additional information regarding the amount of nutrient reduction is appropriate. This information will be included in the revised manuscript, most likely in Table format.

Line 183 effects Line 335, I would not say the decrease in hypoxic duration was large except in a few specific instances. Perhaps moderate would be a better word?

Yes, we agree that "moderate" is a more appropriate description here.

Line 483 'made a first order assessment' might sound a bit less casual than 'took a first order look'

Yes, we agree that this is a better word choice.

Table A2, no caption, chapter 4 needs to be defined here. Table A3, Total, A,B,C and D need to be defined in the caption.

We are impressed with how thoroughly the reviewer read even the appendices! We will definitely fix the "Chapter 4" typo, and define the regions in the caption.

---

## Author Response (AR1)

**Referee comments are shown in blue. Author responses to the Referee's comments are included in black.**

**Anonymous Referee #1**

This manuscript reports on the effect of reduced nutrient loads and climate change scenarios on the degree of hypoxia in Chesapeake Bay using watershed and estuarine models. Such studies can rapidly blow out in complexity and become very hard to communicate concisely. However, the authors skilfully undertook a very nice study that that was well written and structured and broke down some of the likely effects of climate change on hypoxia and compared this to improvements from nutrient load reductions. I found these questions very pertinent based on my recent interactions with coastal management authorities and I think this manuscript will probably a very well cited one in the field.

The model has already been published and was slightly modified for the purposes of this study. I think it was adequately referenced and described.

It was particularly interesting to note that sea level rise could lead to a slight increase in ventilation, and hence increase in DO. As expected, however the overwhelming effect was the reduction in O2 solubility. I also found it interesting and important that the nutrient reductions proved to be worthwhile in the face of climate change (this is a novel and important part of the study). Such modelling exercises have a high degree of uncertainty, but I think this was well addressed in the discussion. The figures were well presented and struck a good balance between clarity and the overwhelming amount of data that such a study invariably produces. It was a pleasure to read such a well written and organised manuscript on a complex topic. I have only some very minor editorial comments.

We would like to thank Anonymous Reviewer #1 for their positive assessment of our manuscript. We agree that it was a challenge to communicate these complex results concisely; this was certainly our goal, and we are pleased to hear that Anonymous Referee #1 thought we succeeded in this regard.

A little more information on the amount of nutrient reduction occurring for the TMDL scenarios would be useful. % reduction is fine.

Yes, we agree that additional information regarding the amount of nutrient reduction is appropriate. This information has been added to Figure 2, and the percents are now given in the text at the end of Section 2.2:

"*Compared to the Base run, the TMDL scenarios include a Bay-wide reduction in riverine nutrient loading of 45%, 44% and 38% for the three years (1993 to 1995) respectively (Fig. 2a).*"

Line 183 effects

Thank you for noticing this typo. This sentence has been corrected/modified.

Line 335, I would not say the decrease in hypoxic duration was large except in a few specific instances. Perhaps moderate would be a better word?

Yes, we agree that "moderate" is a more appropriate description here. This has been changed.

Line 483 'made a first order assessment' might sound a bit less casual than 'took a first order look'

Yes, we thank the reviewer for this improved word choice. This change has been made.

Table A2, no caption, chapter 4 needs to be defined here. Table A3, Total, A,B,C and D need to be defined in the caption.

We are very impressed with how thoroughly the reviewer read the manuscript! We apologize for these typos and omissions. They have been corrected.

**Anonymous Referee #2**

We would like to thank Anonymous Reviewer #2 for their review of our manuscript, and are glad to hear that they found the manuscript to be well written. As Reviewer #1 noted, it is a challenge to present such complex results (with so many variables changing in time and x,y,z space) in a concise manner.

The reviewer's comments, however, make us feel that perhaps the overall objectives of our study may not have been clear. Our study is structured as an initial exploration of the potential ramifications of the *first order* impacts of climate change on oxygen concentrations in the Chesapeake Bay. In the Chesapeake Bay region, many researchers, ourselves included, are working on continuous, long-term ~50-year simulations that include all possible climate effects: changes in solar radiation, humidity, and winds for example, in addition to the other effects examined here (changes in temperature, precipitation and sea level rise). This initial study, however, opts for a different and less complex approach, whereby the impacts of a few first order factors are studied in detail via sensitivity analysis in an effort to help inform future studies. We by no means have meant to imply that we are predicting what the Bay will look like in 2050. We leave this to future work that is focusing on incorporating all climate change effects simultaneously, and running realistic 50+year simulations.

In the first sentence of our "*Methodological Limitations*" section we explicitly state: "*This research is a first order look at the potential impacts that changes in climate may have*

*on the efficacy of nutrient reduction efforts in the Chesapeake Bay; however, more robust examinations of the problem are needed in order to adequately aid in the regulatory decision making process going forward.*" We also end this section by saying: "*To address these limitations, an effort to conduct a continuous simulation from 2015 – 2050 including both gradual implementation of the nutrient reductions and climate change impacts is currently underway.*" However the reviewer's comments indicate to us that this clearly needs to be explained up front in the introduction as well. Thus, for the next version of our manuscript we are making modifications to our introduction to make our objectives for this analysis, and our future work, clearer for future readers.

We appreciate the opportunity to address each of the reviewer's specific comments below.

*1) The simulations did not consider the impact of temperature changes on hydrodynamics. Wind and evaporation did not change be definition. Only the impacts of increases in air temperature, global sea level, river flow and nutrient loads (related to river flow) were considered. Hence, the approach is not dynamically consistent.*

As described in Section 2.3.1, air temperature was not changed in our sensitivity studies. Instead, water temperature was changed consistently throughout the water column. This choice is rationalized in the second paragraph of section 2.3.1, where we cite prior studies that have documented that surface and bottom waters of the Bay are warming uniformly and thus stratification in the Bay is not being substantially impacted by these warmer temperatures. The significant impact of future temperatures on continental shelf and open ocean stratification is well known; however previous studies have indicated that this will likely not be a significant effect in the much shallower and relatively well-mixed Chesapeake Bay. Again, here we are only looking at first-order climate change impacts on DO – the impact of changes in temperature on solubility and growth/grazing/remineralization dynamics. Future work will look at the possibility of second-order effects due to warming-induced hydrodynamic changes. This has now been made clearer in the revised text at the end of Section 2.3.1.

In addition, in the abstract and largely throughout the text, we have modified our text to indicate more clearly that we are examining how mid-21st century projected changes may impact Chesapeake Bay hypoxia, rather than stating that we are running a "2050 scenario" (which, among other things, would require knowledge of wind patterns in 2050.) We believe this will help make sure our future readers are not similarly confused with regards to the overarching goals of our study.

*2) A time slice approach was chosen and the transient behavior was neglected. A period of only three years was investigated. Uncertainty caused by natural variability was not investigated. In particular, the impacts of the large variability of sea level pressure and wind fields on the simulation results were not considered because the time slices are too short. Hence, it is not clear to me whether the calculated changes in dissolved oxygen concentrations are statistically significant.*

A "time slice" approach typically involves running a simulation at a future time interval, for example 2046-2050, rather than for a complete long-term simulation, e.g. 1990-2050. Here we opt for neither of these approaches, but instead adopt a third approach that involves looking at the sensitivity of a simulation to environmental changes. Our four-year simulations (one year spin-up, three year simulation) are not meant to be representative of 2046-2050. Instead we hold winds, humidity and solar radiation constant in order to look at the sensitivity to first-order environmental impacts. Thanks to the reviewer's comments, we feel this this point has now been made clearer in our methods section. In particular, the title of section 2.3 has been changed from "2050 Climate Change Scenarios" to "Climate Change Sensitivity Experiments" and throughout the text we have changed references to "2050" to "mid-21$^{st}$ century" to make it clearer that our estimated impacts are meant to be rough estimates for what will happen in the middle of this century, and not exactly what the conditions will be in 2046-2050.

Also we note that natural interannual variability in Chesapeake Bay hypoxia is overwhelmingly dominated to first order by whether a particular year is characterized by higher than average rainfall (a "wet" year) or lower than average rainfall (a "dry" year). Here we carefully investigate both very wet and very dry years (see Section 2.2, including new modifications). We do document differences in results for the two "types" of years (recall our finding that a wet year with the TMDL nutrient reductions has more hypoxia than a dry year without the nutrient reductions), but generally our primary conclusions hold regardless of whether a year is particularly wet or dry.

*3) The applied changes in air temperature, sea level and river flow were estimated from ensemble mean values from global model simulations from the literature (partly grey literature) and are not consistent results of changing climate in the region. In particular, it was impossible for me to understand how the watershed simulations were done. A regional climate model with sufficient horizontal resolution was not applied. Hence, the simulations are not dynamically consistent projections. I suggest to call them sensitivity studies.*

Yes, we absolutely agree that throughout the manuscript we are performing "sensitivity studies" and are not providing "projections" for 2050. This is an important change that has now been made throughout the manuscript text, specifically where we describe the paper's objectives. We appreciate the reviewer pointing out this source of confusion.

We feel that our estimates of future changes in temperature (+1.75°C), SLR (0.5m) and riverine loading (Fig. 2a) are indeed consistent with peer-reviewed estimates (as described in Section 2.3.) Because the goal of this study is to provide a first order assessment of the sensitivity of hypoxia to potential future changes in temperature, SLR and riverine loading, we feel that applying a regional climate model is outside the scope of this paper. However, work towards this goal is underway, and has been greatly facilitated by the results of these preliminary sensitivity experiments described in this manuscript.

We also apologize that the source of our estimates of future river flow were not clear. In this case we feel that shortening this section to make it more equivalent to the temperature and SLR sections is the best option. As in the case of temperature (Section 2.3.1) and SLR (2.3.2) here we are primarily stating the estimates we've made and briefly describing where these estimates came from. For temperature and SLR these estimates come from peer-reviewed publications (largely Muhling et al., 2017; Ding and Elmore, 2015; Boon and Mitchell, 2015). Estimates of future river run-off have been provided to us by the CBP. Specifically, extensive effort by the CBP over the past year has lead to a report on the Climate Change Assessment Framework for the Watershed Model: http://www.chesapeake.org/stac/presentations/279_CCAF_STACPeerReviewDocument ation_Draft_063017.pdf. This report has also recently been reviewed by independent experts on the Scientific and Technical Advisory Committee panel: http://www.chesapeake.org/pubs/386_Herrmann2018.pdf. Just as we do not describe the detailed methodologies used by Muhling et al. (2017), Ding and Elmore (2015) and Boon and Mitchell (2015), we feel it is outside of the scope of this paper to describe in detail the methodology imposed by the CBP, as this information is openly available online (see above links, for example). We hope that shortening this section will also help make it clearer to our readers that we are indeed performing sensitivity experiments, and not trying to project what dissolved oxygen concentrations will actually be in 2050.

*4) The uncertainty of projected future climate caused by biases of global climate models was not assessed. Usually there is a large spread of projected changes around the ensemble mean. The spread of the calculated changes in dissolved oxygen concentrations may be larger than the differences between the impact of nutrient load changes and the impact of climate change on the results. Hence, the conclusions on the competing impacts may not hold in a multi-model ensemble approach.*

It is true that we did not assess the impacts of biases in the Global Climate Models, but this is because we are performing a sensitivity study, examining how sensitive Chesapeake Bay oxygen concentrations are to changes in water temperature, sea level rise and river flow.

Again, the goal of this research, which we believe is now clearer in the revised version of the manuscript, is to provide a first look at the sensitivity of oxygen concentrations and hypoxic volume to these three environmental forcing changes. Ideally the sensitivity of the estuary would be tested for a number of different temperature changes that would encompass our uncertainties in future temperatures. This work is underway, but is beyond the scope of this preliminary assessment. An additional analysis that involves examining the increase in temperature required to completely nullify all positive impacts of the TMDL reduction (for DO < 5 mg/L) could be conducted. (Recall from Figure 4 that an increase of 1.75°C results in a 40% reduction in the gains of the TMDL. A temperature of 3 or 4°C might result in a complete negation of all TMDL gains.) Because our results show hypoxia is not very sensitive to changes in SLR and changes in river inputs, it is far less critical to examine how sensitive hypoxia is to varying levels of SLR and river inputs.

*5) A greenhouse gas emission scenario RCP 4.5 was chosen. The question whether the conclusions would also hold for RCP 8.5, which is not necessarily less likely than RCP 4.5, was not addressed.*

Although our estimates of future change are broadly consistent with RCP4.5 assumptions, they are not very different from what we would expect for RCP8.5. A number of studies, e.g., Goberville et al. (2015), have demonstrated that the difference between RCP scenarios is smaller than the spread of individual global climate models that utilize the RCP emission scenarios.

In the revised version of the manuscript, all reference to RCP4.5 has been removed. This was done to ensure that our readers fully understand that we are running sensitivity experiments to understand the impact of warming, SLR and precipitation changes on hypoxia in Chesapeake Bay, and not performing specific RCP scenario analyses. We believe this has clarified the manuscript considerably, and thank the reviewer for pointing out this potential source of confusion.

*6) Quantitative figures of changing nutrient concentrations and nutrient supply are not given. Hence, it is difficult to compare with other coastal seas with comparable environmental situation.*

This is an excellent point that was also alluded to by Reviewer #1. The revised manuscript contains a new Figure (Fig. 2a) that shows the total Bay-wide riverine nitrogen loading entering the Bay for each sensitivity experiment. We appreciate the referee noting this omission.

*7) Why has sea level rise a positive impact on dissolved oxygen concentrations in regions B and C? This result is unexpected. The authors state that the impact of increased stratification and residence time is smaller than the impact of increased estuarine circulation. Is this result supported by other model studies or possibly a short-coming of the present model?*

The positive impact of SLR on bottom DO concentrations (Figure 3) was surprising to us as well, though we were reassured when presentations by the Chesapeake Bay Program's modeling team showed the same result for their preliminary Mid-Point Assessment simulations (e.g. https://www.chesapeakebay.net/channel_files/25275/purpose_of_wqstm_overview_6-5-17.pdf). Because these WQSTM simulations have not appeared in a peer-reviewed publication yet (and only exist in the gray literature), we don't feel it's appropriate to cite them here, but they do give us some confidence in our results.

We also feel that we must do a better job of explaining this result in the manuscript, so we have rewritten Section 4.2. As we discuss there, rising sea level increases estuarine circulation and thus reduces residence time in the bottom water and consequently increases bottom oxygen concentrations. But this impact is primarily at the bottom of the water column in the deepest portions of the Bay, as we now illustrate in our new Table 4. At higher oxygen concentrations (3 < DO < 5 mg L$^{-1}$) we see that the increased estuarine circulation (and resulting increased stratification which has been documented by a number of other studies as cited in the manuscript) actually reduces oxygen concentrations. We feel this is a very interesting result of our analysis (because of the complexities and competing impacts described above) and we hope our new Table 4 and revised discussion helps future readers to better understand these results. This finding will also help direct future research questions regarding the important impact of SLR on DO in the Chesapeake Bay.

*In summary, the present study is not about changing climate with all its uncertainties and the approach does not support the provided conclusions that otherwise would have large impact on marine management. Hence, I recommend to perform longer simulations to estimate uncertainties caused by natural variability (usually 30-year long simulations are recommended to address the statistics of weather). To estimate uncertainties due to model biases an ensemble of simulations driven by various global model results should be performed. Further, a high-end emission scenario like RCP 8.5 should be investigated to be able to conclude (perhaps) that the impact of changing climate does not counteract the impact of nutrient load reductions. I recommend a major revision.*

We completely agree with the reviewer that the present study is not about changing climate with all its uncertainties. We hope this is now clearer to our readers in our revised manuscript, and we apologize for the confusion in our initial draft. We feel that some of our terminology (e.g. "scenarios" rather than "sensitivity experiments") was the source of this confusion, and this has been improved in the new draft based on this reviewer's comments.

We hope it is now clear that this study aims to present results of sensitivity studies which illustrate the relative impact that three first order environmental variables may have on Chesapeake Bay oxygen concentrations in the future: water temperature, sea level rise, and river flow. As discussed in our "*Methodological Limitations*" section, the work described here is being followed by a larger study involving experts in global climate models and downscaling techniques – just as the reviewer recommends. This future study will indeed involve longer-term simulations (1985-2050), address uncertainties in climate model biases, and directly include changes in humidity, solar radiation, and winds, in addition to the variables investigated here. Nevertheless, we feel that the results we present here are robust and worthy of publication as they have clearly established several new results that have not been published before, namely that: (1) the potential impacts of near-term climate change on DO will be significantly smaller than improvements in DO expected in response to the required nutrient reductions, especially at the anoxic and hypoxic levels, and (2) increased temperature exhibits the strongest control on the change in future DO concentrations, while sea level rise and increased winter river flow have considerably smaller impacts. This is a particularly crucial result that is very pertinent to future studies that involve actual 2050 scenarios: future work should focus primarily on temperature impacts, since this is likely to be the largest impact on Chesapeake Bay hypoxia.

We feel that our revisions in response to the reviewer's comments have made this a much stronger manuscript, and thus we are very appreciative of the time spent reviewing this manuscript.

[revised manuscript text omitted]